# Social determinants in the delay of starting colorectal cancer treatment

Amanda Almeida Gomes Dantas[1], Nayara Priscila Dantas de Oliveira[2],
Luís Felipe Leite Martins[3], Marianna de Camargo Cancela[4],
Junior Smith Torres-Roman[5], Dyego Leandro Bezerra de Souza[6]*

1 Graduate Program in Health Sciences, Federal University of Rio Grande do Norte – UFRN, Natal, Rio Grande do Norte, Brazil, 2 Department of Physical Therapy, University of Pernambuco– UPE, Petrolina, Pernambuco, Brazil, 3 Surveillance and Situation Analysis Division, Prevention and Surveillance Coordination (CONPREV), National Cancer Institute (INCA), Ministry of Health, Rio de Janeiro, Rio de Janeiro, Brazil, 4 Graduate Program in Oncology, Research and Innovation Coordination, National Cancer Institute (INCA), Ministry of Health, Rio de Janeiro, Rio de Janeiro, Brazil, 5 Universidad Científica del Sur, Lima, Peru, 6 Graduate Program in Public Health, Federal University of Rio Grande do Norte – UFRN, Natal, Rio Grande do Norte, Brazil

* dyego.souza@ufrn.br

## Abstract

Delayed access to colorectal cancer (CRC) treatment leads to disease progression, advanced stages at diagnosis, and reduced survival. In Brazil, Law 12.732/2012 mandates treatment initiation within 60 days of diagnosis; however, disparities persist. To assess compliance with the 60-day treatment initiation window for CRC in Brazil and its association with individual, socioeconomic, and healthcare system factors. We conducted a longitudinal observational retrospective cohort study using data from 65,582 CRC cases diagnosed between 2013 and 2019 in Brazil. Individual-level data were obtained from the Integrator of Hospital Cancer Registries, while socioeconomic and healthcare infrastructure data were extracted from national databases. A Multilevel Poisson Regression model with a random intercept was applied. Approximately forty-two percent of patients experienced treatment delays beyond 60 days, with significant variability among Federal Units (UFs). The highest delay rates were observed in Goiás (60.7%), Pará (60.6%), and Acre (58.4%). Factors associated with delayed treatment included younger age, non-white race, low educational level, rectal tumor location, referral through the Unified Health System (SUS), and receiving treatment outside the patient's municipality of residence. Social determinants significantly impact CRC treatment delays in Brazil, highlighting the need to strengthen healthcare policies to ensure equitable and timely access to cancer treatment, ultimately improving patient survival.

**Data availability statement:** The primary aggregated data used in this study were extracted from official websites open for public consultation, which are available at the following links: https://www.ibge.gov.br/https://irhc.inca.gov.br/RHCNet/http://www.atlasbrasil.org.br/https://datasus.saude.gov.br/cnes-equipes-de-saudehttps://datasus.saude.gov.br/cnes-recursos-humanos-a-partir-de-agosto-de-2007-ocupacoes-classificadas-pela-cbo-2002https://datasus.saude.gov.br/acesso-a-informacao/producao-ambulatorial-sia-sus/ This study used aggregated data from secondary sources, obtained from official publicly accessible platforms, with no possibility of individual identification. All data were fully anonymized prior to access. Thus, in accordance with the Ethics Statement, the data used comply with the guidelines and regulations of the Research Ethics Committee (CEP), which exempts submission to the system and waives the requirement for informed consent, in accordance with Resolution 510/2016 currently in force in Brazil24. The database that was organized from the primary data to support the results of this study is available from the : https://doi.org/10.34740/kaggle/dsv/12930429.

**Funding:** ALGD received doctoral scholarship funding from the Coordination for the Improvement of Higher Education Personnel (CAPES) – Brazil – Code 001. DLBS acknowledges support from the Brazilian National Council for Scientific and Technological Development (CNPq), Productivity Grant No. 308168/2020-8. The funders had no role in study design, data collection and analysis, decision to publish, or preparation of the manuscript.

**Competing interests:** The authors declare no competing interests.

## Introduction

Cancer remains a major global health challenge, with approximately 20 million new cases and 10 million deaths reported in 2022. Demographic projections indicate that the annual incidence is expected to reach 35 million cases by 2050, representing a 75% increase. Among specific cancer types, colorectal cancer (CRC) was the third most common incident 9.6/100,000 cases, second leading cause of cancer-related mortality, with 9.3 cases/100,000 [1]. In Brazil, the National Cancer Institute (INCA) estimates an incidence rate of 11.43 cases per 100,000 inhabitants for the 2023–2025 period, ranking CRC as the fourth most frequent cancer, while the mortality rate registered in 2021 was 9.6, placing it third [2,3].

In this context, CRC represents a significant health issue, with survival directly related to the stage at diagnosis and timely access to treatment [4]. Studies show that diagnostic delays longer than 30 days are associated with worse five-year overall survival [5], and that each four-week delay in starting treatment increases the risk of mortality by 13% to 15%, potentially reaching 39% with a twelve-week delay [6]

Ensuring timely access to therapy is therefore a critical priority for improving CRC prognosis and reducing the burden on health systems. The social determinants of health including economic, political, cultural, and environmental factors play a crucial role in shaping individual health outcomes. These determinants significantly influence the development and progression of CRC by affecting access to healthcare, early detection, and treatment [7]. Furthermore, the social and community context in which individuals are born and live contributes to health disparities across chronic diseases, including cancer [8].

This impact becomes evident in the context of cancer treatment, where structural and social barriers hinder effective access to care. The shortage of infrastructure, specialists, and essential medications, combined with lack of reimbursement and high costs, reflects economic and political inequalities that compromise service delivery [9] Additionally, cultural factors such as fear of treatment side effects and preference for traditional practices, along with low health literacy and misinformation, contribute to treatment non-adherence especially in low- and middle-income countries like Brazil [10,11].

Thus, understanding treatment delays requires a comprehensive analysis of the patient's journey, which involves several stages: symptom onset, initial contact with healthcare services, diagnostic evaluation, referral to a specialist, and initiation of treatment [12,13]. Each of these stages represents a potential source of delay, particularly within public health systems such. Given these challenges, this study aims to evaluate the adequacy of the time interval for CRC treatment initiation in Brazil, as mandated by Law 12.732/2012, and its association with individual, contextual, socioeconomic, and healthcare system factors. By identifying the key determinants of delayed treatment, this research seeks to inform public policies aimed at improving equitable access to timely cancer care.

## Methods

A longitudinal observational retrospective cohort study was conducted using secondary data obtained from the Brazilian Cancer Hospital Registry Integrator (RHC). This

system contains standardized information on sociodemographic characteristics, hospital care activities, and tumor clinical features of patients with malignant neoplasms treated in oncology-accredited specialized services. Although the RHC does not actively follow individuals over time, patients are passively monitored throughout their oncology care pathways, and the continuous recording of events enables the establishment of the temporal relationship between exposures and outcomes [14].

The RHC was established in 1983 by the Brazilian National Cancer Institute (INCA) and was nationally consolidated in 2007 with the creation of the IntegradorRHC system. This registry is responsible for the continuous, systematic, and standardized collection of clinical and epidemiological data on patients with malignant neoplasms treated in public, private, philanthropic, and university hospitals across Brazil. Data collection is conducted using standardized instruments such as the Tumor Registration Form and Follow-up Form, applying internationally recognized coding systems (ICD-10 and ICD-O-3) and the SisRHC software, which includes automated routines for data verification and validation. The periodic submission of information to INCA is mandatory for oncology-accredited hospitals within the Brazilian Unified Health System (SUS), including High-Complexity Oncology Care Centers (CACONs) and High-Complexity Oncology Care Units (UNACONs), as established by Ministerial Ordinances No. 3.535/1998 and No. 741/2005, and voluntary for other institutions [14].

The RHC system follows rigorous procedures for quality control and data consistency, guided by indicators defined in official INCA manuals, such as variable completeness, internal consistency, and time intervals between diagnosis and treatment. It also implements auditing and validation routines before national database consolidation. These standards and indicators are publicly available on INCA's official website, ensuring transparency, traceability, and scientific reliability of the data, as well as methodological robustness, standardization, and sound governance for studies that rely on this information [14].

The study included cases of malignant neoplasms of colon cancer (C18), rectosigmoid junction (C19), and rectum (C20), classified according to the 10th revision of the International Classification of Diseases (ICD-10) [15]. Eligible individuals were aged 18–99 years, diagnosed between 2013 and 2019, and monitored in hospital oncology care services in Brazil.

Excluded were cases from this study if they had missing information on age, sex, place of residence, date of diagnosis, or date of treatment initiation; cases with prior diagnosis and treatment; cases with missing data on prior diagnosis and treatment; and those who did not undergo initial treatment due to refusal, advanced disease, or death; Cases that underwent treatment at a different hospital unit and received a different form of treatment, cases that did not receive any type of treatment and had no information on treatment, and cases of carcinoma in situ (TNM 0).

The primary outcome was the time to treatment initiation, as defined by Law 12.732/2012, which mandates that cancer patients receiving care through the SUS begin treatment within 60 days of diagnosis [16]. The outcome was dichotomized into ≤ 60 days (on-time treatment) and > 60 days (delayed treatment).

The RHC data were collected by Federative Unit (UF) of residence of the cases and linked to other databases aggregated by UF. Sociodemographic variables were collected from the Atlas of Human Development in Brazil, such as the Gini Index, available from the United Nations Development Programme (UNDP) [17], and the Vulnerability Index from the Institute for Applied Economic Research (IPEA), all referring to the year 2010 [18].

Data on the density of healthcare professionals and the availability of healthcare services were obtained from the National Registry of Health Establishments (CNES) [19] and the Health Outpatient Information System (SIA) [20]. From these data, indicators were calculated for the years 2014 and 2017: "Density of Family Health Physicians" (Number of family physicians per 100.000 inhabitants); "Density of Coloproctologists" (Number of coloproctologists per 1.000.000 inhabitants); "Density of Oncologists" (Number of oncologists per 1.000.000 inhabitants); "Density of Surgical Oncologists" (Number of surgical oncologists per 1.000.000 inhabitants); "Density of Oncology Services" (Number of accredited oncology services per 1.000.000 inhabitants); "Density of Oncology Beds" (Number of oncology beds per 100.000 inhabitants); "Density of Radiotherapy Equipment" (Number of radiotherapy equipment per 100,000 inhabitants).

The independent variables (Fig 1) were organized according to the theoretical model of the Social Determinants of Health (SDH) proposed by the Commission on Social Determinants of Health (CSDH) of the World Health Organization (WHO). This conceptual framework includes three main categories of SDH: sociopolitical context, socioeconomic position, and intermediate health determinants [21].

In the sociopolitical dimension, variables reflecting the socioeconomic conditions of Brazilian states were considered, such as the Gini Index and the Social Vulnerability Index (SVI). In the structural determinants, individual-level variables representing the socioeconomic position of patients with CRC were included, such as educational level. Finally, the intermediate determinants comprised biological and behavioral factors, along with local indicators related to the availability of and access to health services, such as age group and oncology service density. Fig 1 illustrates the distribution of these variables according to the aforementioned theoretical model.

For the calculation of the indicators, the 2010 Demographic Census was used, and population counts and estimates by UF, sex, and age were obtained from the Brazilian Institute of Geography and Statistics (IBGE) [22] for the respective years.

The cancer care network in Brazil integrates actions from both the public and private sectors, covering all stages from prevention to palliative care. This system encompasses Primary Health Care (PHC), home care, and specialized outpatient and hospital services, supported by technical, normative, logistical, and governance structures that ensure coordination and effectiveness of care [23]. Primary Health Care, through basic health units and Family Health Strategy teams, serves as the main entry point to the SUS and is responsible for screening and referring suspected cases to specialized care [24]. Specialized care, in turn, is provided by various units such as polyclinics, hospital outpatient clinics, regional reference centers, and autonomous services linked to the private sector. These services rely on specialized professionals who confirm diagnoses and monitor patients' clinical progress, in addition to performing specific tests essential for both diagnosis and treatment monitoring [25].

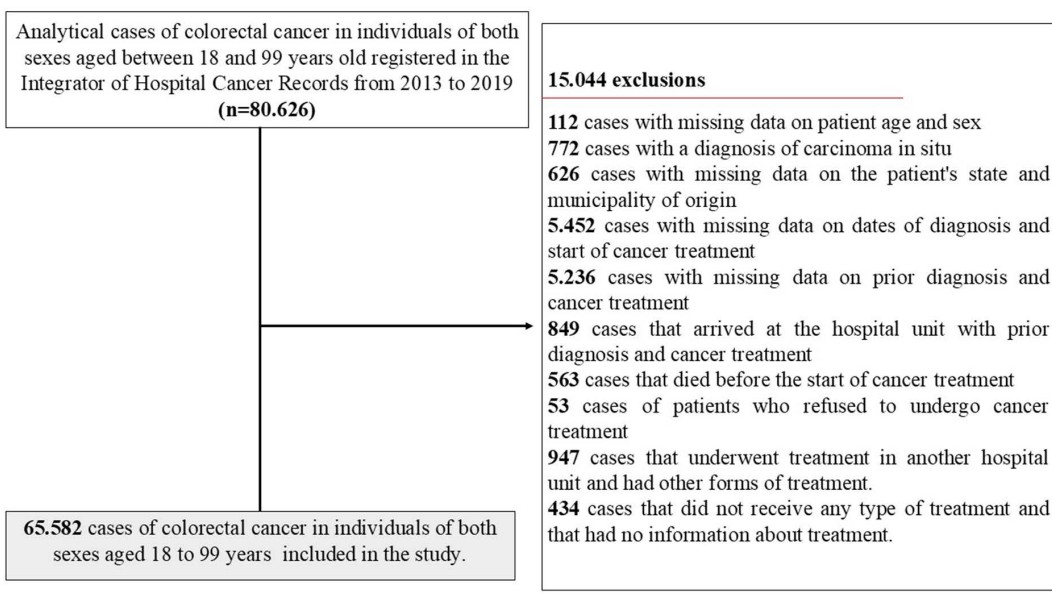

**Fig 1. Theoretical explanatory model of the social determinants of health related to delays in the initiation of colorectal cancer treatment in Brazil.** This figure was developed by the authors for the present study, based on the conceptual framework proposed by the World Health Organization Commission on Social Determinants of Health, and does not reproduce or adapt copyrighted material. The figure is published under the Creative Commons Attribution License (CC BY 4.0).

 

Cancer treatment is multimodal, encompassing surgical interventions, radiotherapy, and chemotherapy, and is characterized by high costs. It is performed in specialized units organized into different levels of complexity. Notable examples include: (1) CACONs, which provide treatment for all types of cancer, including hematological ones, and may or may not offer pediatric care; (2) UNACONs, which focus on managing the most prevalent cancer types, with or without radiotherapy, hematology-oncology, and/or pediatric oncology services; and (3) hospital complexes composed of general hospitals that perform oncologic surgeries and provide radiotherapy services, acting as complementary units affiliated with CACONs or UNACONs [26].

In this context, it is important to highlight that CACONs offer a wide range of services, including specialized medical consultations. This structure allows patients who seek these centers directly, without prior referral from other levels of the healthcare network, to receive diagnostic confirmation and initiate treatment more promptly. Conversely, those who obtain a diagnosis in specialized care units, such as polyclinics or hospital outpatient clinics, may experience longer waiting times before the actual start of treatment [26].

For this reason, the analyses between groups were conducted separately, distinguishing individuals without treatment at the time of admission who arrived with a prior diagnosis obtained elsewhere in the healthcare network from those who were also without treatment and entered directly into high-complexity hospitals, where they received both diagnosis and treatment within the same services.

Data analysis was conducted using descriptive statistics, summary measures, and graphical representations. Quantitative variables were categorized into tertiles or as dichotomous variables (categorized using the median) when necessary for bivariate and multilevel analysis.

A spatial analysis of the data was conducted using georeferencing with Geoda 1.6 software, utilizing Ufs to create thematic maps. This analysis spatially describes the distribution of the proportion of delay in the initiation of colorectal cancer treatment in Brazil for the period from 2013 to 2019.

The maps were developed based on the territorial cartographic mesh (shapefiles), publicly provided by IBGE, without copyright restrictions. The cartographic meshes can be accessed at: https://www.ibge.gov.br/geociencias/organizacao-do-terr.

To assess the association between the dependent and independent variables, Pearson's Chi-Square test was performed. Since the evaluated outcome has a prevalence greater than 10% and contextual variables were used, a crude multivariate analysis was conducted using a Poisson Regression model with robust variance. Subsequently, a Multilevel Poisson Regression model with a random intercept was applied, based on the results of the Likelihood Ratio (LR) test.

The empty model, with a random intercept, was initially evaluated. Next, individual-level variables were added, maintaining the random intercept, and the reduction in the variability of the random effect was examined by comparing it with the previous model. Then, contextual-level variables were included in the modeling. Variables that remained in the final model were those that were statistically significant according to the Wald Test ($\alpha = 0.05$), as well as those with theoretical plausibility for inclusion in the final statistical model [27].

To determine the dose-response effect between the study's independent variables and the proportion of delay in initiating CRC treatment, the Trend Test (p-trend) was performed for eligible variables included in the final multilevel model [27]. These analyses were conducted using Stata 15.1 software.

Additionally, overdispersion was assessed by examining model residuals and performing specific tests. The overdispersion test was conducted using the goodness-of-fit statistic based on Pearson's chi-square. For the analysis of individuals who arrived with a diagnosis but without treatment, the obtained value was 23,871.48 with 41,828 degrees of freedom, resulting in a dispersion index of approximately 0.57. For those who arrived without diagnosis and without treatment, the test value was 6,892.39 with 41,828 degrees of freedom, resulting in a dispersion index of about 0.16. Since the dispersion indices for both tests were below 1, this indicates an absence of overdispersion in the analyzed data.

Although the outcome refers to an event that occurs over time (treatment initiation), it was operationalized as a binary variable (>60 days) rather than as a time-to-event measure. Thus, the proportion of delayed cases at the time of analysis reflects a prevalent condition. In this context, PR were estimated using Poisson Regression with robust variance, which provides unbiased and more interpretable measures of association than odds ratios in studies with high outcome frequency, and avoids the inflation of associations commonly observed with logistic regression [28,29].

This study used aggregated data from secondary sources, obtained from official publicly accessible platforms, with no possibility of individual identification. All data were fully anonymized prior to access. Thus, in accordance with the Ethics Statement, the data used comply with the guidelines and regulations of the Research Ethics Committee (CEP), which exempts submission to the system and waives the requirement for informed consent, in accordance with Resolution 510/2016 currently in force in Brazil [30].

## Results

Between 2013 and 2019, a total of 80,626 colorectal cancer cases were recorded in individuals aged 18–99 years. After exclusions, 37.975 (57.9%) accessed treatment within 60 days, while 27.607 (42.1%) experienced a delay in treatment initiation (>60 days after diagnosis) including all cases of people who arrived with and without a diagnosis (Table 1).

According to the study's eligibility criteria, 65,582 cases of CRC cancer from the country's UFs were included, as shown in Fig 2.

Fig 3 spatially presents the distribution of the proportions of delays in initiating CRC treatment across all 26 Brazilian UFs and the Federal District. The proportion of delays in CRC treatment initiation in Brazil from 2013 to 2019 was 42.1% (95% CI: 41.7–42.4), including all cases of people who arrived with and without a diagnosis, varying across states and regions of the country.

The highest proportions of delays in treatment initiation for this condition are located in the Central-West and North regions, with emphasis on Goiás (60.7%), Pará (60.6%), and Acre (58.4%).

The majority of cases with delays in treatment initiation were men, aged 60–69 years, non-white, and illiterate or with incomplete primary education, as shown in Table 1.

The unadjusted prevalences and PR for delays in treatment initiation will be presented for individuals who arrived with a diagnosis but without treatment and those without a diagnosis and without treatment in Tables 2 and 3.

Statistically significant associations were observed between delays in initiating CRC treatment for individuals who arrived with and without a diagnosis and all individual, socioeconomic contextual variables, and healthcare service availability, as shown in Tables 2 and 3.

Tables 4 and 5 show the results of the multilevel analysis of the data. The multilevel modeling, through the initial empty model, provides statistical evidence that the variation among the Brazilian UFs is different from zero, according to the Likelihood Ratio (LR) test for CRC among those who arrived with a diagnosis but without treatment (LR 396.65; p < 0.001) and without a diagnosis and without treatment (LR 20.96; p < 0.001).

The results of the multilevel analysis for CRC indicate that the delay in starting treatment for individuals who arrived with diagnosis but without treatment is associated with younger age groups (PR 0.86; 95% CI 0.82–0.89) compared to the older age group, non-white race (PR 1.07; 95% CI 1.03–1.12), low education (PR 1.13; 95% CI 1.10–1.17), referral by the public health system (SUS) (PR 1.25; 95%CI 1.18–1.32) and when the treatment hospital is not located in the patient's municipality of residence (PR 1.10; 95%CI 1.07–1.13). The contextual level variables were not associated with the investigated outcome, as shown in Table 4.

The results of the multilevel analysis for individuals who arrived without a diagnosis and without treatment indicate that delays in initiating CRC treatment are associated with younger age groups (PR 0.80; 95% CI 0.72–0.89) compared to the oldest age group, low education level (PR 1.16; 95% CI 1.08–1.26), tumor location in the rectum (PR 1.53; 95% CI 1.43–1.64), and referral through the SUS (PR 1.70; 95% CI 1.45–1.98). Contextual-level variables were not associated with the investigated outcome, as shown in Table 5.

**Table 1. Descriptive analysis of time to initiation of colorectal cancer treatment according to individual characteristics and contextual variables. Brazil, by state of residence (n = 65,582).**

| | ≤ 60 days | | >60 days | |
|---|---|---|---|---|
| | n | % | n | % |
| **Individual variables** | | | | |
| **Sex** | | | | |
| Female | 19.078 | 57.6 | 14.030 | 42.4 |
| Male | 18.897 | 58.1 | 13.577 | 41.9 |
| **Age Group** | | | | |
| 18–49 years old | 7.111 | 63.1 | 4.149 | 36.8 |
| 50–59 years old | 9.044 | 57.9 | 6.577 | 42.1 |
| 60–69 years old | 11.200 | 56.5 | 8.594 | 43.4 |
| 70 years or older | 10.620 | 56.1 | 8.287 | 43.8 |
| **Race** | | | | |
| White | 13.795 | 63.1 | 8.041 | 36.8 |
| Non white | 11.136 | 58.0 | 8.063 | 42.0 |
| No Information | 13.044 | 53.1 | 11.503 | 46.8 |
| **Education** | | | | |
| None/Incomplete fundamental education | 20.216 | 55.1 | 16.435 | 44.8 |
| Fundamental education | 6.974 | 60.8 | 4.485 | 39.1 |
| Secondary education/ Incomplete | 10.785 | 61.7 | 6.687 | 38.2 |
| **Tumor Location** | | | | |
| Colon | 20.307 | 59.3 | 13.938 | 40.7 |
| Rectal | 17.668 | 56.3 | 13.669 | 43.7 |
| **Stage** | | | | |
| TNM 1 | 8.218 | 57.8 | 6.001 | 42.2 |
| TNM 2 | 9.512 | 58.8 | 6.654 | 41.1 |
| TNM 3 | 3.807 | 57.3 | 2.832 | 42.6 |
| TNM 4 | 5.997 | 55.2 | 4.850 | 44.7 |
| No Information | 10.441 | 58.9 | 7.270 | 41.0 |
| **First Treatment Received** | | | | |
| Surgery | 26.403 | 65.0 | 14.162 | 34.9 |
| Radiotherapy | 4.883 | 65.7 | 2.543 | 34.2 |
| Chemotherapy | 6.585 | 38.0 | 10.741 | 61.9 |
| Hormone therapy/Immunotherapy | 97 | 58.4 | 69 | 41.5 |
| Others | 7 | 7.0 | 92 | 92.9 |
| **Diagnosis and Prior Treatment** | | | | |
| No diagnosis and no treatment | 20.010 | 84.2 | 3.734 | 15.7 |
| With diagnosis and without treatment | 17.965 | 42.9 | 23.873 | 57.0 |
| **Reference Source** | | | | |
| Public (SUS) | 18.632 | 58.6 | 13.146 | 41.3 |
| Private/Insurance | 4.111 | 68.7 | 1.865 | 31.2 |
| No Information | 15.232 | 54.7 | 12.596 | 45.2 |
| **Hospital Location** | | | | |
| It is in the municipality of origin | 18.027 | 54.6 | 11.036 | 45.3 |
| Not located in the municipality of origin | 19.948 | 62.0 | 16.571 | 37.9 |

*(Continued)*

**Table 1.** (Continued)

| | ≤ 60 days | | >60 days | |
|---|---|---|---|---|
| | n | % | n | % |
| **Socioeconomic contextual variables** | | | | |
| **Gini Index** | | | | |
| 0.450–0.560 | 29.176 | 58.7 | 20.470 | 41.2 |
| 0.590–0.650 | 8.799 | 55.2 | 7.137 | 44.7 |
| **Social Vulnerability Index (IVS)** | | | | |
| 0.192–0.277 | 17.575 | 62.8 | 10.410 | 37.2 |
| 0.289–0.297 | 11.542 | 54.2 | 9.728 | 45.7 |
| 0.319–0.521 | 8.858 | 54.2 | 7.469 | 45.7 |
| **Health service offer contextual variables** | | | | |
| **Density of Family Health Physicians (per 100,000 inhabitants)** | | | | |
| 11.51–16.08 | 11.849 | 53.7 | 10.202 | 46.2 |
| 17.04–19.09 | 13.728 | 61.5 | 8.565 | 38.4 |
| 20.33–33.96 | 12.398 | 58.3 | 8.840 | 41.6 |
| **Density of Oncologist (per 100,000 inhabitants)** | | | | |
| 5.72–29.55 | 13.525 | 58.7 | 9.485 | 41.2 |
| 29.90–40.92 | 17.426 | 56.8 | 13.247 | 43.1 |
| 44.45–47.70 | 7.024 | 59.0 | 4.875 | 42.1 |
| **Density of Coloproctologist (per 1000,000 inhabitants)** | | | | |
| 0.00–1.76 | 14.623 | 60.5 | 9.512 | 39.4 |
| 1.81–1.87 | 11.596 | 54.5 | 9.646 | 45.4 |
| 2.08–10.40 | 11.756 | 58.1 | 8.449 | 41.8 |
| **Density of Surgical Oncologists (per 1000,000 inhabitants)** | | | | |
| 0.00–3.39 | 16.033 | 68.2 | 7.454 | 31.7 |
| 3.62–6.50 | 13.793 | 62.6 | 8.222 | 37.3 |
| 8.77–14.47 | 12.012 | 59.8 | 8.068 | 40.1 |
| **Density of Oncology Services (per 100,000 inhabitants)** | | | | |
| 1.93–5.24 | 13.548 | 57.6 | 9.939 | 42.3 |
| 5.27–7.71 | 12.024 | 54.6 | 9.991 | 45.3 |
| 7.81–14.23 | 12.403 | 61.7 | 7.677 | 38.2 |
| **Density of Oncology Beds (per 100,000 inhabitants)** | | | | |
| 0.65–4.33 | 4.641 | 55.5 | 11.704 | 44.4 |
| 4.41–10.81 | 12.681 | 65.2 | 6.749 | 34.7 |
| 13.96–13.96 | 10.653 | 53.7 | 9.154 | 46.2 |
| **Density of Radiotherapy Equipment (per 100,000 inhabitants)** | | | | |
| 0.00–3.31 | 21,169 | 54.4 | 17.723 | 45.5 |
| 3.68–3.68 | 6,573 | 59.0 | 4.564 | 40.9 |
| 3.87–4.85 | 10,233 | 65.7 | 5.320 | 34.2 |

## Discussion

The findings indicate that individual-level factors including age, race, education level, tumor location, treatment hospital, and reference source were significant determinants for the delay in access to CRC treatment. In contrast, contextual-level variables did not show a statistically significant association. Overall, the proportion of treatment delay was 42.1%. When stratified by diagnostic status, the delay was substantially higher among individuals who arrived with a diagnosis

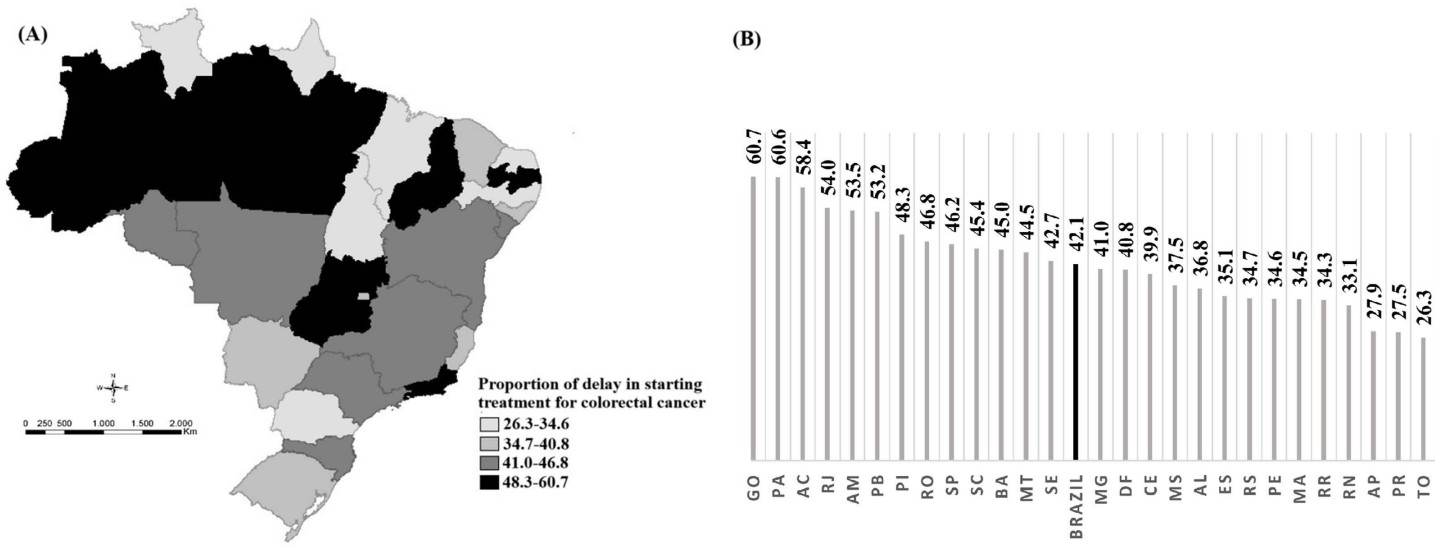

**Fig 2. Flowchart describing the selection process of colorectal cancer cases from 2013 to 2019 in the Cancer Hospital Registry Integrator.**

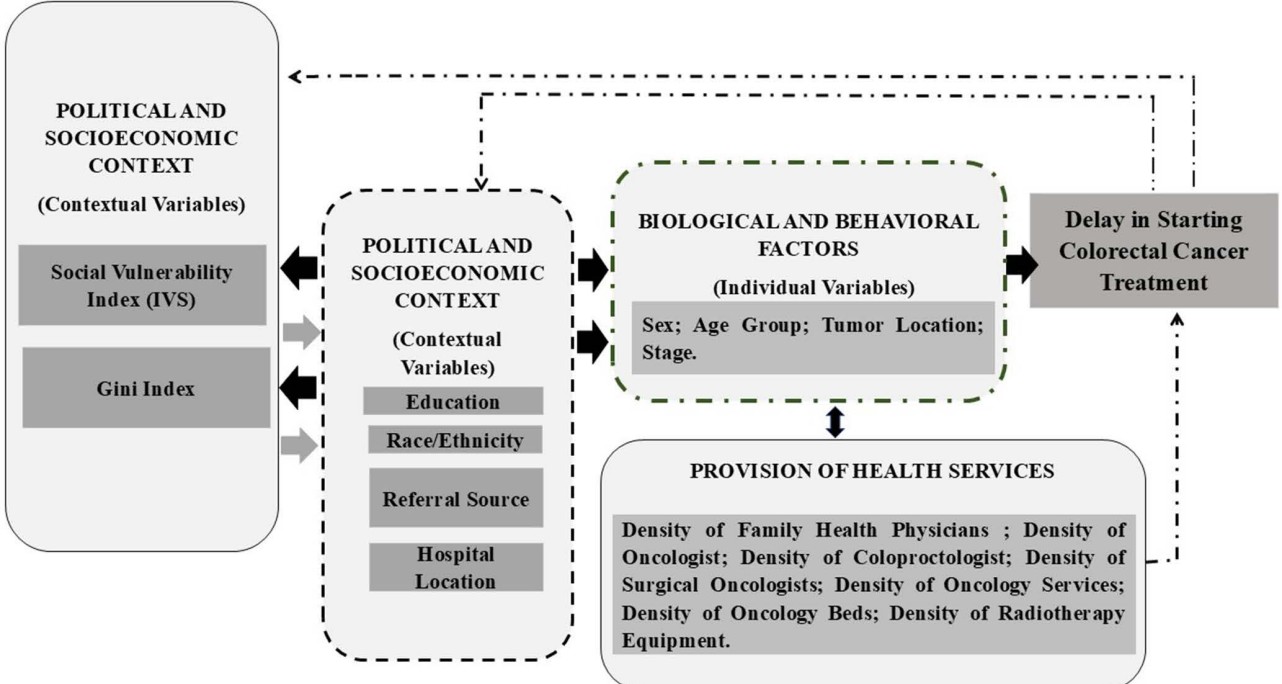

**Fig 3. Spatial distribution of the proportion of delays in initiating colorectal cancer treatment from 2013 to 2019, by Federative Unit (UF) (n = 65,582).**

**Table 2. Prevalence and unadjusted prevalence ratios for delay in the initiation of colorectal cancer treatment among diagnosed but untreated individuals, according to individual characteristics and contextual variables. Brazil, by state of residence (n=41,838).**

Delay in Initiating Colorectal
Cancer Treatment

| | n | % | PR | CI (95%) | p* |
|---|---|---|---|---|---|
| **Individual Variables** | | | | | |
| **Sex** | | | | | |
| Female | 11.705 | 57.5 | 1.02 | 0.99-1.03 | 0.07 |
| Male | 12.168 | 56.6 | 1.00 | – | |
| **Age Group** | | | | | |
| 18–49 years old | 3.639 | 49.8 | 0.84 | 0.82-0.87 | |
| 50–59 years old | 5.719 | 55.8 | 0.95 | 0.93-0.97 | <0.001* |
| 60–69 years old | 7.426 | 58.7 | 1.00 | – | |
| 70 years or older | 7.089 | 60.8 | 1.03 | 1.01-1.05 | |
| **Race** | | | | | |
| White | 6.991 | 51.2 | 1.00 | – | < 0.001* |
| Non white | 7.062 | 56.2 | 1.07 | 1.03-1.11 | |
| No Information | 9.820 | 62.7 | 1.11 | 1.08-1.14 | |
| **Education** | | | | | |
| None/Incomplete fundamental education | 14.333 | 60.0 | 1.12 | 1.10-1.14 | <0.001* |
| Fundamental education | 3.896 | 52.7 | 0.98 | 0.96-1.01 | |
| Secondary education/ Incomplete | 5.644 | 53.4 | 1.00 | – | |
| **Stage** | | | | | |
| TNM 1 | 5.232 | 55.8 | 0.97 | 0.95-0.99 | |
| TNM 2 | 5.784 | 54.8 | 0.95 | 0.93-098 | <0.001* |
| TNM 3 | 2.471 | 59.1 | 1.03 | 1.00- 1.06 | |
| TNM 4 | 4.192 | 60.2 | 1.05 | 1.02-1.07 | |
| No Information | 6.194 | 57.3 | 1.00 | – | |
| **Tumor Location** | | | | | |
| Colon | 12.156 | 59.6 | 1.00 | – | |
| Rectal | 11.717 | 54.6 | 1.12 | 1.10- 1.15 | <0.001* |
| **First Treatment Received** | | | | | |
| Surgery | 12.082 | 52.7 | 6.48 | 3.17-13.23 | <0.001* |
| Radiotherapy | 2.244 | 46.6 | 7.32 | 3.58-14.94 | |
| Chemotherapy | 9.395 | 67.7 | 4.42 | 2.16- 9.03 | |
| Hormone therapy/Immunotherapy | 63 | 46.6 | 7.31 | 3.52-15.18 | |
| Others | 89 | 92.7 | 1.00 | – | |
| **Reference Source** | | | | | |
| Public (SUS) | 11.514 | 55.4 | 1.21 | 1.17-1.26 | <0.001* |
| Private/Insurance | 1.678 | 45.4 | 1.00 | – | |
| No Information | 10.681 | 61.5 | 1.35 | 1.30-1.40 | |
| **Hospital Location** | | | | | |
| It is in the municipality of origin | 14.843 | 59.1 | 1.00 | – | |
| Not located in the municipality of origin | 9.030 | 53.9 | 1.09 | 1.07-1.11 | <0.001* |
| **Socioeconomic contextual variables** | | | | | |
| **Gini Index** | | | | | |
| 0.450-0.560 | 17.692 | 56.3 | 1.00 | – | |
| 0.590-0.650 | 6.181 | 59.2 | 1.05 | 1.03-1.07 | <0.001* |

*(Continued)*

**Table 2.** (Continued)

| Delay in Initiating Colorectal Cancer Treatment | n | % | PR | CI (95%) | p* |
|---|---|---|---|---|---|
| **Social Vulnerability Index (IVS)** | | | | | |
| 0.192-0.277 | 9.169 | 51.5 | 1.00 | – | |
| 0.289-0.297 | 8.165 | 62.3 | 1.21 | 1.18-1.23 | <0.001* |
| 0.319-0.521 | 6.539 | 59.7 | 1.15 | 1.13-1.18 | |
| **Health service offer contextual variables** | | | | | |
| **Density of Family Health Physicians (per 100,000 inhabitants)** | | | | | |
| 11.51-16.08 | 5.110 | 37.2 | 1.23 | 1.19-1.23 | <0.001* |
| 17.04-19.09 | 6.143 | 45.8 | 1.22 | 1.19-1.23 | |
| 20.33-33.96 | 6.712 | 45.6 | 1.00 | – | |
| **Density of Oncologist (per 100,000 inhabitants)** | | | | | |
| 5.72-29.55 | 8.293 | 55.0 | 1.00 | 0.98-1.03 | |
| 29.90-40.92 | 11.251 | 59.6 | 1.08 | 1.06-1.11 | <0.001* |
| 44.45-47.70 | 4.329 | 54.7 | 1.00 | – | |
| **Density of Coloproctologist (per 1000,000 inhabitants)** | | | | | |
| 0.00-1.76 | 8.565 | 52.9 | 0.92 | 0.92-0.94 | |
| 1.81-1.87 | 8.089 | 61.8 | 1.07 | 1.05-1.09 | <0.001* |
| 2.08-10.40 | 7.219 | 57.0 | 1.00 | | |
| **Density of Surgical Oncologists (per 1000,000 inhabitants)** | | | | | |
| 0.00-3.39 | 10.073 | 55.1 | 1.02 | 0.99-1.05 | <0.001* |
| 3.62-6.50 | 10.282 | 60.3 | 1.11 | 1.09-1.14 | |
| 8.77-14.47 | 3.518 | 53.8 | 1.00 | – | |
| **Density of Oncology Services (per 100,000 inhabitants)** | | | | | |
| 1.93-5.24 | 8.914 | 55.6 | 1.02 | 1.00-1.04 | <0.001* |
| 5.27-7.71 | 8.445 | 61.2 | 1.12 | 1.10-1.15 | |
| 7.81-14.23 | 6.514 | 54.2 | 1.00 | – | |
| **Density of Oncology Beds (per 100,000 inhabitants)** | | | | | |
| 0.65-4.33 | 10.390 | 57.9 | 0.92 | 0.90-0.94 | <0.001* |
| 4.41-10.81 | 5.770 | 49.6 | 0.79 | 0.71-0.82 | |
| 13.96-13.96 | 7.713 | 62.6 | 1.00 | – | |
| **Density of Radiotherapy Equipment (per 100,000 inhabitants)** | | | | | |
| 0.00-3.31 | 15.186 | 60.6 | 1.22 | 1.20-1.25 | <0.001* |
| 3.68-3.68 | 4.104 | 54.5 | 1.10 | 1.07-1.13 | |
| 3.87-4.85 | 4.583 | 49.4 | 1,.00 | – | |

PR: Estimated Prevalence Ratio by Robust Poisson Model; CI: Confidence interval; p: Wald's test; *Statistically significant.

but without treatment (57.1%) compared to those who arrived without a diagnosis and without treatment (15.7%). These delays also varied significantly across different states and regions.

This result is particularly relevant because, given the way the Brazilian health system is organized especially under the framework of the Network for Chronic Disease Care and the National Policy for Oncology Care it would be expected that a prior diagnosis would facilitate and expedite the start of treatment. However, the findings reveal the opposite: patients diagnosed in other points of the healthcare network face greater barriers to timely access to

**Table 3. Prevalence and unadjusted prevalence ratios for delay in the initiation of colorectal cancer treatment among individuals who arrived without diagnosis and without treatment, according to individual characteristics and contextual variables. Brazil, by state of residence (n = 23,744).**

| Delay in Initiating Colorectal Cancer Treatment | n | % | PR | CI (95%) | p* |
|---|---|---|---|---|---|
| **Individual Variables** | | | | | |
| **Sex** | | | | | |
| Female | 1.862 | 16.0 | 0.96 | 0.90-1.02 | 0.20 |
| Male | 1.872 | 15.4 | 1.00 | – | |
| **Age Group** | | | | | |
| 18–49 years old | 510 | 12.8 | 0.78 | 0.71-0.86 | |
| 50–59 years old | 858 | 15.9 | 0.97 | 0.89-1.05 | <0.001* |
| 60–69 years old | 1.168 | 16.3 | 1.00 | – | |
| 70 years or older | 1.198 | 16.5 | 1.00 | 0.93-1.08 | |
| **Race** | | | | | |
| White | 1.050 | 12.8 | 1.00 | – | < 0.001* |
| Non white | 1.001 | 15.0 | 1.17 | 1.14-1.38 | |
| No Information | 1.683 | 18.9 | 1.47 | 1.37-1.58 | |
| **Education** | | | | | |
| None/Incomplete fundamental education | 2.102 | 16.4 | 1.09 | 1.01-1.16 | <0.001* |
| Fundamental education | 589 | 14.4 | 0.95 | 0.87-1.05 | |
| Secondary education/ Incomplete | 1.043 | 15.1 | 1.00 | – | |
| **Stage** | | | | | |
| TNM 1 | 769 | 15.8 | 0.99 | 0.98-1.01 | |
| TNM 2 | 870 | 15.4 | 1.00 | 0.98-1.01 | <0.001* |
| TNM 3 | 361 | 14.6 | 1.01 | 0.99-1.03 | |
| TNM 4 | 658 | 16.9 | 0.98 | 0.96-1.00 | |
| No Information | 1.076 | 15.5 | 1.00 | – | |
| **Tumor Location** | | | | | |
| Colon | 1.782 | 12.8 | 1.00 | – | |
| Rectal | 1.952 | 19.7 | 1.53 | 1.44–1.63 | <0.001* |
| **First Treatment Received** | | | | | |
| Surgery | 2.080 | 11.7 | 0.11 | 0.11-0.12 | <0.001* |
| Radiotherapy | 299 | 11.4 | 0.11 | 0.10-0.12 | |
| Chemotherapy | 1.346 | 38.9 | 0.38 | 0.37-0.40 | |
| Hormone therapy/Immunotherapy | 6 | 19.3 | 0.19 | 0.09-0.39 | |
| Others | 3 | 10.0 | 1.00 | – | |
| **Reference Source** | | | | | |
| Public (SUS) | 1.632 | 14.8 | 1.81 | 1.56-2.09 | <0.001* |
| Private/Insurance | 187 | 8.1 | 1.00 | – | |
| No Information | 1.915 | 18.3 | 2.23 | 1.93-2.58 | |
| **Hospital Location** | | | | | |
| It is in the municipality of origin | 1.728 | 15.1 | 1.00 | – | |
| Not located in the municipality of origin | 2.006 | 16.3 | 0.92 | 0.87-0.98 | <0.001* |
| **Socioeconomic contextual variables** | | | | | |
| **Gini Index** | | | | | |
| 0.450-0.560 | 2.778 | 15.2 | 1.00 | – | |

*(Continued)*

**Table 3.** (Continued)

| Delay in Initiating Colorectal Cancer Treatment | n | % | PR | CI (95%) | p* |
|---|---|---|---|---|---|
| 0.590-0.650 | 956 | 17.3 | 1.14 | 1.06-1.22 | <0.001* |
| **Social Vulnerability Index (IVS)** | | | | | |
| 0.192-0.277 | 1.241 | 12.1 | 1.00 | – | |
| 0.289-0.297 | 1.563 | 19.1 | 1.56 | 1.46-1.58 | <0.001* |
| 0.319-0.521 | 930 | 17.3 | 1.42 | 1.31-1.53 | |
| **Health service offer contextual variables** | | | | | |
| **Density of Family Health Physicians (per 100,000 inhabitants)** | | | | | |
| 11.51-16.08 | 1.588 | 19.0 | 1.48 | 1.37-1.60 | <0.001* |
| 17.04-19.09 | 1.306 | 14.6 | 1.14 | 1.05-1.23 | |
| 20.33-33.96 | 840 | 12.8 | 1.00 | – | |
| **Density of Oncologist (per 100,000 inhabitants)** | | | | | |
| 5.72-29.55 | 1.192 | 15.0 | 1.09 | 0.99-1.20 | |
| 29.90-40.92 | 1.996 | 16.9 | 1.23 | 1.13-1.34 | <0.001* |
| 44.45-47.70 | 546 | 13.6 | 1.00 | – | |
| **Density of Coloproctologist (per 1000,000 inhabitants)** | | | | | |
| 0.00-1.76 | 947 | 11.9 | 0.73 | 0.68-0.80 | |
| 1.81-1.87 | 1.557 | 19.1 | 1.18 | 1.10-1.26 | <0.001* |
| 2.08-10.40 | 1.230 | 16.1 | 1.00 | | |
| **Density of Surgical Oncologists (per 1000,000 inhabitants)** | | | | | |
| 0.00-3.39 | 1.123 | 14.0 | 0.94 | 0.86-1.02 | <0.001* |
| 3.62-6.50 | 1.907 | 17.3 | 1.16 | 1.07-1.26 | |
| 8.77-14.47 | 704 | 14.8 | 1.00 | – | |
| **Density of Oncology Services (per 100,000 inhabitants)** | | | | | |
| 1.93-5.24 | 1.025 | 13.7 | 0.95 | 0.88-1.03 | <0,001* |
| 5.27-7.71 | 1.546 | 18.8 | 1.30 | 1.21-1.39 | |
| 7.81-14.23 | 1.163 | 14.4 | 1.00 | – | |
| **Density of Oncology Beds (per 100,000 inhabitants)** | | | | | |
| 0.65-4.33 | 1.314 | 15.5 | 0.81 | 0.75-0.86 | <0.001* |
| 4.41-10.81 | 979 | 12.5 | 0.65 | 0.60-0.70 | |
| 13.96-13.96 | 1.441 | 19.2 | 1.00 | – | |
| **Density of Radiotherapy Equipment (per 100,000 inhabitants)** | | | | | |
| 0.00-3.31 | 2.537 | 18.3 | 1.55 | 1.44-1.68 | <0.001* |
| 3.68-3.68 | 460 | 12.7 | 1.08 | 0.97-1.21 | |
| 3.87-4.85 | 737 | 11.7 | 1.00 | – | |

PR: Estimated Prevalence Ratio by Robust Poisson Model; CI: Confidence interval; p: Wald's test; *Statistically significant.

therapy, highlighting the fragmentation and lack of coordination among levels of care. This scenario suggests failures in communication and care regulation, undermining the comprehensiveness of care and the effectiveness of oncology care pathways [23–26].

A high proportion of treatment delay, similar to what was observed in this study, has also been reported in other parts of the world. In Indonesia, for example, more than half of the patients experienced delays exceeding 180 days before initiating treatment [31]. Similarly, in Canada, one-quarter of the population analyzed had an average waiting time of up to

**Table 4. Multilevel analysis of individual characteristics and contextual variables for delay in initiating colorectal cancer treatment among individuals aged 18 to 99 years diagnosed but not treated between 2013 and 2019. Brazil, by state of residence (n = 41,838).**

| Variables | Empty model | Model 1 | | Model 2 | |
|---|---|---|---|---|---|
| | | PR (CI 95%) | *p* | PR (CI 95%) | *p* |
| **Level 1 (Individual)** | | | | | |
| **Sex** | | | | | |
| Female | – | 1.02 (0.99–1.04) | 0.11 | 1.02 (0.99–1.04) | 0.11 |
| Male | – | 1.00 | | 1.00 | |
| **Age Group** | | | | | |
| 18–49 years old | – | 0.86 (0.82–0.89) | | 0.86 (0.82–0.89) | |
| 50–59 years old | – | 0.95 (0.92–0.98) | | 0.95 (0.92–0.98) | |
| 60–69 years old | – | 1.00 | <0.001* | 1.00 | <0.001* |
| 70 years or older | – | 1.03 (0.99–1.06) | | 1.03 (0.99–1.06) | |
| **Race** | | | | | |
| White | – | 1.00 | | 1.00 | |
| Non white | – | 1.07 (1.03-1.12) | <0.001* | 1.07 (1.03–1.12) | <0.001* |
| No Information | – | 1.18 (1.11-1.25) | | 1.18 (1.11–1.25) | |
| **Education** | | | | | |
| None/Incomplete fundamental education | – | 1.13 (1.10–1.17) | <0.001* | 1.13 (1.10–1.17) | <0.001* |
| Fundamental education | – | 1.03 (0.99–1.07) | | 1.03 (0.99–1.07) | |
| Secondary education/ Incomplete | – | 1.00 | | 1.00 | |
| **Stage** | | | | | |
| TNM 1 | – | 0.95 (0.91–0.99) | | 0.95 (0.91–0.99) | |
| TNM 2 | – | 0.94 (0.91–0.98) | <0.001* | 0.94 (0.91–0.98) | <0.001* |
| TNM 3 | – | 1.02 (0.97–1.08) | | 1.02 (0.97–1.08) | |
| TNM 4 | – | 1.00 (0.96–1.04) | | 1.00 (0.96–1.04) | |
| No Information | – | 1.00 | | 1.00 | |
| **Reference Source** | | | | | |
| Public (SUS) | – | 1.25 (1.18–1.32) | <0.001* | 1.25 (1.18–1.32) | <0.001* |
| Private/Insurance | – | 1.00 | | 1.00 | |
| No Information | – | 1.26 (1.18–1.34) | | 1.26 (1.18–1.34) | |
| **Hospital Location** | | | | | |
| It is in the municipality of origin | – | 1.00 | <0.001* | 1.00 | <0.001* |
| Not located in the municipality of origin | – | 1.10 (1.07–1.13) | | 1.10 (1.07–1.13) | |
| **Level 2 (Contextual)** | | | | | |
| **Gini Index** | | | | | |
| 0.450-0.560 | – | – | – | 1.00 | 0.96 |
| 0.590-0.650 | – | – | | 1.00 (0.66–1.52) | |
| **Density of Oncology Services (per 100,000 inhabitants)** | | | | | |
| 1.93-5.24 | – | – | – | 0.93 (0.60-1.44) | 0.55 |
| 5.27-7.71 | – | – | – | 0.74 (0.43–1.27) | |
| 7.81-14.23 | – | – | – | 1.00 | |
| **Fixed effects** | – | | | | |
| Intercept (CI 95%) | −0.586 (−0.646—0.525) | 0.382 (0.349–0.418) | | 0.367 (0.320–0.421) | |
| **Random effects** | | | | | |
| Variance (CI 95%) | 0.021 (0.011–0.038) | 0.018 (0.010–0.034) | | 0.017 (0.009–0.032) | |

*(Continued)*

**Table 4.** (Continued)

| Variables | Empty model | Model 1 | | Model 2 | |
|---|---|---|---|---|---|
| | | PR (CI 95%) | *p* | PR (CI 95%) | *p* |
| LR Test (x². p-value) | 396.65 (<0.001) | 298.93 (<0.001) | | 234.01 (<0.001) | |

PR: Prevalence ratio adjusted by the multilevel model with random intercept; CI: Confidence interval; p: Wald's test; * *p trend* ≤ 0.05.

**Model 1:** Statistical model with inclusion of individual level variables; **Model 2:** Statistical model with inclusion of variables of individual level and contextual level per FU.

218 days between diagnosis and treatment [32], while in the Netherlands, the reported average delay was 138 days [33]. These findings illustrate that treatment delays are a global challenge, affecting both low- and high-income countries.

The delay in initiating CRC treatment, when compared across Brazilian federal units in this study and based on the theoretical model of the SDH, was associated, at the level of intermediate determinants, with younger age groups as well as with tumor location in the rectum (biological factor). Regarding socioeconomic position, associations were observed with low educational level, race, location of the treatment hospital, and referral source. No significant associations were found in the sociopolitical context or in the intermediate context related to access to health services.

In this study, younger patients both with and without a diagnosis were more likely to initiate treatment within the timeframe established by law, compared to older individuals. This finding may be explained by the more aggressive pathological features commonly observed in younger patients with colorectal cancer, such as poorly differentiated adenocarcinoma, signet ring cell carcinoma, and evidence of lymphovascular or perineural invasion [34]. The clinical complexity associated with these tumor characteristics often leads to earlier recognition and prioritization of treatment, as supported by previous studies [35,36].

Recent epidemiological studies have consistently shown a concerning increase in the incidence of early-onset colorectal cancer (EO-CRC), defined as cancer diagnosed in individuals younger than 50 years [37,38]. This upward trend is observed globally and here in Brazil, as demonstrated in this study, and represents a significant public health challenge, since EO-CRC patients often present with advanced-stage disease and face delays in diagnosis. In response, organizations such as the USPSTF and NCCN have revised their guidelines, lowering the recommended starting age for screening from 50 to 45 years [39,40]. These changes highlight the importance of including younger populations in screening programs an initiative that should be incorporated in Brazil when a specific program for this condition is implemented within the SUS, aiming to align public health policies with the current scenario.

Delays in initiating cancer treatment are known to negatively impact clinical outcomes, leading to worse prognoses, increased mortality, and higher treatment complexity [40,41]. These delays disproportionately tend to disproportionately affect vulnerable populations, such as patients from disadvantaged racial and ethnic groups and those residing in resource-poor and high-deprivation areas [40–42]. Such disparities are often rooted in structural and systemic barriers, including limited access to specialized oncology services, long referral pathways, insufficient health infrastructure, and socioeconomic constraints. These conditions are commonly observed in public health systems in low- and middle-income countries, where patients depend heavily on fragmented care networks [43–45]. In the context of this study, the association between delayed treatment and social vulnerability underscores the urgent need for interventions focused on equity in cancer care delivery. Strategies such as strengthening primary care referral systems, expanding regional oncology services, and implementing patient navigation programs may help reduce delays and mitigate long-standing disparities in access to timely treatment.

Our findings corroborate this reality, revealing a greater delay in access to treatment for non-white patients who arrived with a diagnosis. Even with the availability of adequate treatment for most patients with CRC, individuals of other ethnicities experience less access to consultations and delays in essential treatments, such as surgery, chemotherapy, and

**Table 5. Multilevel analysis of individual characteristics and contextual variables for delay in initiating colorectal cancer treatment among individuals aged 18 to 99 years undiagnosed and untreated between 2013 and 2019. Brazil, by state of residence (n = 23,744).**

| Variables | Empty model | Model 1 | | Model 2 | |
|---|---|---|---|---|---|
| | | PR (CI 95%) | p | PR (CI 95%) | p |
| **Level 1 (Individual)** | | | | | |
| **Sex** | | | | | |
| Female | – | 0.98 (0.91–1.04) | 0.55 | 0.98 (0.91–1.04) | 0.55 |
| Male | – | 1.00 | | 1.00 | |
| **Age Group** | | | | | |
| 18–49 years old | – | 0.80 (0.72–0.89) | | 0.80 (0.72–0.89) | |
| 50–59 years old | – | 0.97 (0.89–1.06) | | 0.97 (0.89–1.06) | |
| 60–69 years old | – | 1.00 | <0.001* | 1.00 | <0.001* |
| 70 years or older | – | 1.01 (0.93–1.10) | | 1.01 (0.93–1.10) | |
| **Education** | | | | | |
| None/Incomplete fundamental education | – | 1.16 (1.08–1.26) | <0.001* | 1.16 (1.08–1.26) | <0.001* |
| Fundamental education | – | 1.05 (0.95–1.17) | | 1.05 (0.95–1.17) | |
| Secondary education/ Incomplete | – | 1.00 | | 1.00 | |
| **Stage** | | | | | |
| TNM 1 | – | 0.98 (0.88–1.08) | | 0.98 (0.88–1.08) | |
| TNM 2 | – | 0.97 (0.88–1.07) | 0.67 | 0.97 (0.88–1.07) | 0.67 |
| TNM 3 | – | 0.93 (0.82–1.05) | | 0.93 (0.82–1.05) | |
| TNM 4 | – | 1.02 (0.91–1.04) | | 1.02 (0.91–1.04) | |
| No Information | – | 1.00 | | 1.00 | |
| **Tumor Location** | | | | | |
| Colon | – | 1.00 | | 1.00 | |
| Rectal | – | 1.53 (1.43–1.64) | <0.001* | 1.53 (1.43–1.64) | <0.001* |
| **Reference Source** | | | | | |
| Public (SUS) | – | 1.70 (1.45–1.98) | <0.001* | 1.70 (1.45–1.98) | <0.001* |
| Private/Insurance | – | 1.00 | | 1.00 | |
| No Information | – | 1.64 (1.37–1.95) | | 1.64 (1.37–1.95) | |
| **Level 2 (Contextual)** | | | | | |
| **Social Vulnerability Index (IVS)** | | | | | |
| 0.192-0.277 | – | – | – | 1.00 | 0.09 |
| 0.289-0.297 | – | – | – | 1.13 (0.92–1.38) | |
| 0.319-0.521 | – | – | – | 1.19 (1.01–1.41) | |
| **Density of Oncology Services (per 100,000 inhabitants)** | | | | | |
| 1.93-5.24 | – | – | – | 0.88 (0.74–1.05) | 0.29 |
| 5.27-7.71 | – | – | – | 0.87 (0.73–1.05) | |
| 7.81-14.23 | – | – | – | 1.00 | |
| **Fixed effects** | | | | | |
| Intercept (CI 95%) | −0.166 (−0.194–−0.138) | 0.071 (0.056–0.490) | | 0. 077 (0.053–0.111) | |
| **Random effects** | | | | | |
| Variance (CI 95%) | 0.0021 (0.000–0.006) | 0.152 (0.075–0.309) | | 0.152 (0.075–0.309) | |
| LR Test (x². p-value) | 20.96 (<0.001) | 167.20 (<0.001) | | 110.88 (<0.001) | |

PR: Prevalence ratio adjusted by the multilevel model with random intercept; CI: Confidence interval; p: Wald's test; * p trend ≤ 0.05.

**Model 1:** Statistical model with inclusion of individual level variables; **Model 2:** Statistical model with inclusion of variables of individual level and contextual level per FU.

radiotherapy, which results in more frequent disease recurrence and higher cancer-related mortality when compared to white patients. These racial and ethnic disparities in treatment reflect, in part, underlying socioeconomic disparities between the groups [46,47].

Chow [48] and Zarcos-Pedrin [49] point to low education as a factor that hinders timely access to treatment, which is consistent with the results of this research, in which patients with low levels of education, whether diagnosed or not, tend to experience delays in accessing treatment. This may be a consequence of low health literacy, which can compromise treatment adherence, difficulties in understanding medical instructions, and failure to attend appointments. Furthermore, these individuals may have a greater tendency to avoid information about the disease and develop a pessimistic view regarding cancer [49].

The literature has shown that the time interval for surgical intervention in rectal cancer, when compared to colon cancer, tends to be longer. This disparity possibly stems from the more frequent use of magnetic resonance imaging in rectal tumors. The increased use of this examination can result in delays in the surgical procedure, due to the need for more detailed multidisciplinary consultations and, in some cases, the implementation of neoadjuvant care [50,51]. These factors may explain the results of this research, which indicate a higher prevalence of delay in access to treatment in patients without a diagnosis in rectal cancer.

Additionally, improvements in therapies for various types of cancer have provided an increase in survival across multiple stages of the disease. These advances, combined with other factors, contribute to the reduction of cancer-related mortality. However, significant disparities in access to healthcare persist, preventing the benefits of better outcomes from reaching broad segments of the population [52].

Within this context, geographical disparities emerge as determining factors in health outcomes, particularly concerning cancer. This influence largely stems from unequal access to health services. Patients who reside in remote areas and face greater geographical distances tend to have reduced or limited access to tertiary care, which, in turn, contributes to less favorable health outcomes [53].

Furthermore, we found that patients who sought treatment in hospitals outside their municipality of residence were more likely to experience delays, particularly among those who already had a confirmed diagnosis. This scenario reflects a structural deficiency in the geographic distribution of oncology services and highlights the burden of indirect medical costs such as travel, lodging, and food which disproportionately affect lower-income individuals [54].

This difficulty in accessing treatment is closely tied to the broader issue of inequality in cancer care. Such inequality includes delays in initiating therapy, failure to complete treatment protocols, omission of critical doses, or poor adherence to medical recommendations, all of which increase the risk of tumor progression or recurrence, contribute to worse health outcomes, and result in inefficient use of health system resources [55].

To address these challenges, strategic investment is needed in regional oncology networks, patient transportation programs, and integrated care pathways that minimize the logistical and financial burdens faced by cancer patients. Low- and middle-income countries face substantial challenges in delivering high-quality cancer care compared to high-income nations. These difficulties stem from limitations in the healthcare workforce, scarce financial and technological resources, and restricted access to specialized professionals, which collectively contribute to patients often being diagnosed at advanced or metastatic stages of disease [56,57].

This scenario is consistent with the reality observed in Brazil, where 65.6% of CRC cases were diagnosed at an advanced stage, reflecting persistent difficulties in early detection, timely diagnosis, and prompt treatment factors that directly impact patient survival and treatment outcomes [58].

Despite having the largest universal public health system in the world, Brazil's SUS continues to face structural and organizational barriers that hinder its capacity to respond to complex health needs. While the SUS is mandated to provide comprehensive and free care including health promotion, prevention, diagnosis, treatment, harm reduction, and palliative care across all levels of care (primary, secondary, and tertiary) through an integrated network, systemic deficiencies

remain. These include inequitable resource allocation, fragmented care pathways, and governance challenges, underscoring the need for a strategic redesign of financing, service delivery models, and intersectoral coordination to meet the evolving demands of the Brazilian population and ensure timely, equitable access to cancer care [59,60].

In this context, the fragmentation of referral flows constitutes a significant barrier to the effectiveness of care networks and directly contributes to delays in treatment initiation, especially in conditions requiring rapid response, such as cancer. Many referral records are still made manually or in disconnected systems, which promotes errors and data loss. Additionally, triage and scheduling suffer from long queues, team overload, and the absence of clear clinical criteria, hindering proper prioritization of cases. Communication between services is weakened by lack of feedback, unanswered messages, and the absence of effective tracking mechanisms [61–63]. Logistic problems, such as insufficient transportation, equipment shortages, and geographic barriers, also compromise continuity of care. Added to this is a culture of disorganized handoffs, generating multiple unnecessary referrals. Furthermore, the system's low responsiveness results in "invisible" waiting lists composed of patients who, even after referral, do not receive care, revealing critical failures in coordination and management of care [64,65].

This scenario could explain the findings of this research, in which patients with and without a diagnosis, referred by the public system for treatment, presented a greater delay when compared to patients who arrive through the private service. Furthermore, it is worth noting the absence of a national screening program for CRC, with only local pilot projects in existence [66]. Thus, this gap suggests the lack of care pathways to guide the patient's journey with CRC within the health system, partly explaining the results found here.

Implementing political-level actions to reduce the overall diagnosis interval has been a goal in national health systems in Europe, Australia, Canada, and the United States. To exemplify this effort, the International Cancer Benchmarking Partnership stands out. This international collaboration aims to improve cancer outcomes by comparing and analyzing health data between different countries, including Australia, Canada, Denmark, Norway, Sweden, and the United Kingdom, including delays in diagnosis and treatment [67].

Reorganize the cancer treatment sector in Brazil, it is crucial to promote a comprehensive restructuring, which involves the reorganization of structures, policies and processes, aiming to consolidate better integration of services and coordination of care between providers. Furthermore, it is imperative to implement patient-centered care models adapted to individual needs, overcoming obstacles in accessing healthcare and preventive services. These obstacles include financial impediments such as low personal income and high poverty rates; physical difficulties, such as lack of transportation and limited geographic access to health facilities; and personal barriers, such as cultural and linguistic factors.

This study presents some important limitations that should be considered when interpreting the results. Although the study design is a longitudinal observational retrospective cohort, the outcome was operationalized as a binary variable (>60 days) rather than as a time-to-event measure. Thus, the delay in treatment initiation was analyzed as a prevalent condition, which limits causal inference. Therefore, the associations observed should be interpreted as statistical associations rather than causal effects. In addition, the use of secondary data from Brazil's health information systems represents a methodological limitation, particularly regarding the completeness and quality of the information. Although the RHC stands out as the most comprehensive and systematized source available in the country for cancer diagnosis being essential for epidemiological analyses and widely used in scientific research [68] it is recognized that its data may be subject to errors resulting from manual or incomplete extraction from medical records. Although no formal imputations or sensitivity analyses were performed, this may represent a limitation with potential impact on the findings. Most of the missing data are attributed to gaps in clinical records, which may compromise the quality of some of the variables analyzed.

For this same database, a previous multilevel analysis assessing data completeness demonstrated that missingness was not random but was associated with contextual and individual-level factors. This reinforces the importance of considering potential systematic bias when interpreting our results and highlights the relevance of targeted actions to improve data quality [58].

Additionally, the absence of significant contextual effects in the multilevel models, despite some associations observed in the bivariate analyses, may be related to measurement and aggregation limitations of the contextual variables used. Indicators such as the Gini Index and service density were operationalized at the state level, which may not adequately capture intrastate variability, especially in states with large territorial extension and high socio-spatial heterogeneity. This level of aggregation may obscure internal inequalities and attenuate potentially relevant contextual effects. Furthermore, some indicators may have limited sensitivity or specificity to capture key dimensions related to the outcome studied, which introduces risks of measurement error and ecological fallacy.

Despite these inherent limitations, this study has several strengths that enhance the reliability and generalizability of its findings. It draws on a large, nationally representative dataset derived from the RHC, which operates under standardized data collection, validation, and auditing protocols coordinated by the INCA. The system's adherence to international coding standards (ICD-10 and ICD-O-3) and continuous quality-control routines ensures high data consistency and scientific reliability. Furthermore, the RHC's extensive national coverage and its recognized use in numerous epidemiological and policy-oriented studies reinforce the robustness of the present analyses and their contribution to evidence-based planning and management of cancer care in Brazil [14].

The results found here demonstrate the impact of individual-level social determinants and corroborate the idea that individuals from vulnerable populations, such as patients from certain racial and ethnic groups and unfavorable socioeconomic conditions, face difficulties in accessing oncological care services upon arrival at the SUS, within the 60-day period established by Law 12.732/2012, which delays timely and opportune treatment, regardless of whether they have a prior diagnosis or not. It becomes imperative, therefore, to define the care pathway to be ensured to the patient and to establish the practices and services to be made available by each component of the care network so that they are easily accessible, of quality, and in sufficient quantity.

## Author contributions

**Conceptualization:** Amanda Almeida Gomes Dantas, Marianna de Camargo Cancela, Dyego Leandro Bezerra de Souza.

**Data curation:** Amanda Almeida Gomes Dantas, Nayara Priscila Dantas de Oliveira, Luís Felipe Leite Martins.

**Formal analysis:** Amanda Almeida Gomes Dantas, Nayara Priscila Dantas de Oliveira, Luís Felipe Leite Martins, Junior Smith Torres-Roman.

**Funding acquisition:** Dyego Leandro Bezerra de Souza.

**Methodology:** Amanda Almeida Gomes Dantas, Luís Felipe Leite Martins, Marianna de Camargo Cancela.

**Project administration:** Dyego Leandro Bezerra de Souza.

**Supervision:** Marianna de Camargo Cancela, Dyego Leandro Bezerra de Souza.

**Writing – original draft:** Amanda Almeida Gomes Dantas, Marianna de Camargo Cancela, Junior Smith Torres-Roman, Dyego Leandro Bezerra de Souza.

**Writing – review & editing:** Amanda Almeida Gomes Dantas, Junior Smith Torres-Roman, Dyego Leandro Bezerra de Souza.

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
