## [Decision Letter · Decision Letter 0]

6 Aug 2025

Dear Dr. de Souza,

Thank you for submitting your manuscript to PLOS ONE. After careful consideration, we feel that it has merit but does not fully meet PLOS ONE’s publication criteria as it currently stands. Therefore, we invite you to submit a revised version of the manuscript that addresses the points raised during the review process.

We look forward to receiving your revised manuscript.

Kind regards,

Sreeram V. Ramagopalan

Academic Editor

PLOS ONE

Journal Requirements:

“he author ALGD obtained funding in the doctoral course of the Coordination for the Improvement of Higher Education Personnel - Brazil (CAPES)- Code 001.

The author DLBS thank CNPq (Brazilian National Council for Scientific and Technological Development) productivity grants 308168/2020-8.”

Before we proceed with your manuscript, please address the following prompts.

Reviewers' comments:

Reviewer's Responses to Questions

**Comments to the Author**

1. Is the manuscript technically sound, and do the data support the conclusions?

Reviewer #1: Yes

2. Has the statistical analysis been performed appropriately and rigorously?

Reviewer #1: Yes

3. Have the authors made all data underlying the findings in their manuscript fully available?

Reviewer #1: Yes

4. Is the manuscript presented in an intelligible fashion and written in standard English?

Reviewer #1: Yes

Reviewer #1: Title

Social determinants in the delay of starting colorectal cancer treatment

Reviewer’s Comments

This manuscript explores the influence of individual, socioeconomic, and health system factors on delays in initiating colorectal cancer (CRC) treatment in Brazil, assessing compliance with Law 12.732/2012. Utilizing a robust dataset of 65,582 CRC cases from 2013 to 2019, the authors employed multilevel Poisson regression to identify associations with treatment delays, concluding that significant sociodemographic and systemic disparities affect timely treatment. The manuscript is well written.

Areas of Improvement

Introduction

1. The introduction presents important global cancer statistics, which help contextualize the broader public health relevance of the study. However, given that the focus of the manuscript is specifically on colorectal cancer (CRC), it would strengthen the rationale if the authors included current CRC specific data on incidence and mortality rates, both globally and within Brazil. Furthermore, integrating statistics that explicitly demonstrate how delays in treatment initiation impact CRC outcomes such as stage progression and survival rates would better convey the magnitude and urgency of the problem. This would enhance the reader’s understanding of the study’s significance and the policy implications of its findings.

2. Line 69 pg 4 “ The literature points that delayed treatment is associated with an increase in mortality for all types of cancer”. Authors did a great work trying to emphasize on the impact of delayed in treatment. However, at this stage, the authors have almost provided the conclusion in the introduction. As such, I suggest they remain focused on articulating the problem with CRC treatment in the country. As previously mentioned, authors should consider supporting their claims with figures.

3. While the manuscript presents a compelling analysis of the association between social determinants and delays in CRC treatment initiation, it would benefit from the inclusion of a stronger theoretical framework to contextualize these relationships.

Methods

1. The authors should provide more detailed justification and explain whether any imputation or sensitivity analysis was conducted to address the potential bias caused by missing data and exclusions.

2. The absence of significant contextual effects in the multilevel models, despite some bivariate associations, warrants further discussion. Are there measurement or aggregation issues? Could variables such as the Gini index or service density mask within-state heterogeneity?

3. Although Poisson with robust SE handles variance inflation, authors can enhance their work be clearly stating how overdispersion was considered and managed.

4. Also, it would strengthen the paper if the authors compared results with standard logistic regression models to demonstrate robustness.

Results

1. The U.S. Preventive Services Task Force (USPSTF) recommends routine colorectal cancer (CRC) screening for adults aged 45 to 75 years. Consequently, CRC cases diagnosed in individuals younger than 45 years fall under the category of early-onset colorectal cancer (EO-CRC). In the current study, the authors grouped patients aged 18–49 years together. However, it would be valuable to further disaggregate this group, particularly to identify and report on individuals aged 18–44 who fall below the USPSTF recommended screening threshold. This information could provide important insights into the burden of EO-CRC within the study population. Given the substantial number of cases in this age range, the authors should consider discussing the implications of EO-CRC for screening policies and how it may contribute to delayed diagnosis and treatment initiation in this younger demographic.

2. The tables are well presented but need titles with the total study sample (N=65,582).

3. Please include a scale bar and a north arrow on your map. These elements enhance its readability and professional look. Also, can you include Caption in Figure 2?

Discussion

1. While the discussion mentions structural and systemic causes for disparities, it would benefit from deeper exploration of the mechanisms behind these factors. For example, how does fragmentation in the referral pathway concretely lead to treatment delays? Are there examples of bottlenecks or specific steps in the process where delays are most acute?

Minor comments

1. P.2, ln 41: Please spell out “42.1%” as “Approximately forty-two percent (42.1%) …”

2. You should revise the title for Table 1.

3. Clearly provide titles for all tables and figures.

4. P.44, ln 407: “Referencias” should be “References”

**Do you want your identity to be public for this peer review?** For information about this choice, including consent withdrawal, please see our Privacy Policy

Reviewer #1: No

---

## [Author Response · Author response to Decision Letter 1]

1 Sep 2025

RESPONSE TO REVIEWERS

INTRODUCTION

1.The introduction presents important global cancer statistics, which help contextualize the broader public health relevance of the study. However, given that the focus of the manuscript is specifically on colorectal cancer (CRC), it would strengthen the rationale if the authors included current CRC specific data on incidence and mortality rates, both globally and within Brazil. Furthermore, integrating statistics that explicitly demonstrate how delays in treatment initiation impact CRC outcomes such as stage progression and survival rates would better convey the magnitude and urgency of the problem. This would enhance the reader’s understanding of the study’s significance and the policy implications of its findings.

Dear reviewer, thank you for your valuable contribution. We have accepted your suggestion, as demonstrated below. The corresponding changes have been incorporated into the manuscript and can be found between lines 56 and 65.

• Cancer remains a major global health challenge, with approximately 20 million new cases and 10 million deaths reported in 2022. Demographic projections indicate that the annual incidence is expected to reach 35 million cases by 2050, representing a 77% increase. Among specific cancer types, colorectal cancer (CRC) was the third most incident 9.6 cases, and the second leading cause of cancer-related mortality 9.3¹. In Brazil, the National Cancer Institute (INCA) estimates an incidence rate of 11.43 cases per 100,000 inhabitants for the 2023-2025 period, ranking CRC as the fourth most frequent cancer, while the mortality rate registered in 2021 was 9.6, placing it third², ³.

• . In this context, CRC represents a significant health issue, with survival directly related to the stage at diagnosis and timely access to treatment⁴. Studies show that diagnostic delays longer than 30 days are associated with worse five-year overall survival5, and that each four-week delay in starting treatment increases the risk of mortality by 13% to 15%, potentially reaching 39% with a twelve-week delay6.

2.Line 69 pg 4 “ The literature points that delayed treatment is associated with an increase in mortality for all types of cancer”. Authors did a great work trying to emphasize on the impact of delayed in treatment. However, at this stage, the authors have almost provided the conclusion in the introduction. As such, I suggest they remain focused on articulating the problem with CRC treatment in the country. As previously mentioned, authors should consider supporting their claims with figures.

Dear reviewer, considering your observation, we decided to remove the paragraph in question and include information that further elaborates on the barriers to timely access to cancer treatment, as demonstrated below. Additionally, the numerical data you requested were incorporated into the paragraph, as indicated in Response 1. These changes can be found between lines 74 and 80 of the manuscript.

• This impact becomes evident in the context of cancer treatment, where structural and social barriers hinder effective access to care. The shortage of infrastructure, specialists, and essential medications, combined with lack of reimbursement and high costs, reflects economic and political inequalities that compromise service delivery9 Additionally, cultural factors such as fear of treatment side effects and preference for traditional practices, along with low health literacy and misinformation, contribute to treatment non-adherence especially in low- and middle-income countries like Brazil10-11.

3.While the manuscript presents a compelling analysis of the association between social determinants and delays in CRC treatment initiation, it would benefit from the inclusion of a stronger theoretical framework to contextualize these relationships.

Dear reviewer, thank you for your comment. We would like to inform you that we adopted the theoretical conceptual model of the Social Determinants of Health (SDH), developed by the WHO Commission on Social Determinants of Health, as the basis for structuring the study variables. To enhance clarity, we included an illustrative figure (Figure 1), presented in the Methods section, which represents the organization of the variables according to the model’s three dimensions: sociopolitical context, socioeconomic position, and intermediate health determinants. Additionally, we provided an explanation of how this model was applied in the study. These modifications can be found between lines 130 and 142 of the manuscript.

• The independent variables (Figure 1) were organized according to the theoretical model of the Social Determinants of Health (SDH) proposed by the Commission on Social Determinants of Health (CSDH) of the World Health Organization (WHO). This conceptual framework includes three main categories of SDH: sociopolitical context, socioeconomic position, and intermediate health determinants21.

• In the sociopolitical dimension, variables reflecting the socioeconomic conditions of Brazilian states were considered, such as the Gini Index and the Social Vulnerability Index (SVI). In the structural determinants, individual-level variables representing the socioeconomic position of patients with colorectal cancer (CRC) were included, such as educational level. Finally, the intermediate determinants comprised biological and behavioral factors, along with local indicators related to the availability of and access to health services, such as age group and oncology service density. Figure 1 illustrates the distribution of these variables according to the aforementioned theoretical model.

Finally, this theoretical framework was also incorporated into the discussion section to contextualize and interpret the study’s findings based on the dimensions of the Social Determinants of Health, as demonstrated below. We believe these changes have strengthened the theoretical foundation of the study and contributed to greater methodological clarity and depth in the interpretation of the results. The corresponding modifications can be found between lines 293 and 300 of the manuscript.

• The delay in initiating CRC treatment, when compared across Brazilian federal units in this study and based on the theoretical model of the Social Determinants of Health (SDH), was associated, at the level of intermediate determinants, with younger age groups as well as with tumor location in the rectum (biological factor). Regarding socioeconomic position, associations were observed with low educational level, race, location of the treatment hospital, and referral source. No significant associations were found in the sociopolitical context or in the intermediate context related to access to health services.

METHODS

1. The authors should provide more detailed justification and explain whether any imputation or sensitivity analysis was conducted to address the potential bias caused by missing data and exclusions.

We appreciate the reviewer’s insightful comment. In our analysis, we adopted a complete-case approach, excluding records with missing data. We recognize that this strategy may introduce bias, and have now included a more detailed justification for this methodological choice in the revised manuscript. Although we did not perform formal imputation or sensitivity analyses, we explicitly acknowledge this limitation and discuss its potential impact on our results.

The missing data are primarily due to incomplete or absent information in medical records, which are the source of the secondary data used. This limitation may affect the completeness and overall quality of some variables analyzed. We have added this consideration to the discussion section and emphasized the need for improved data recording practices in future research and health information systems.

Additionally, for this same database, a previous multilevel analysis assessing data completeness demonstrated that missingness was not random, but rather associated with contextual and individual-level factors. This reinforces the importance of accounting for potential systematic bias in the interpretation of our results and highlights the relevance of targeted actions to improve data quality, which was added to the limitations section of the article.

• This study presents some important limitations that should be considered when interpreting the results. The first relates to the cross-sectional design adopted, which prevents the establishment of temporal relationships between the outcomes investigated and the factors analyzed. In addition, the use of secondary data from Brazil’s health information systems represents a methodological limitation, particularly regarding the completeness and quality of the information. Although the RHC stands out as the most comprehensive and systematized source available in the country for cancer diagnosis being essential for epidemiological analyses and widely used in scientific research62 it is recognized that its data may be subject to errors resulting from manual or incomplete extraction from medical records. Although no formal imputations or sensitivity analyses were performed, this may represent a limitation with potential impact on the findings. Most of the missing data are attributed to gaps in clinical records, which may compromise the quality of some of the variables analyzed.

For this same database, a previous multilevel analysis assessing data completeness demonstrated that missingness was not random but was associated with contextual and individual-level factors. This reinforces the importance of considering potential systematic bias when interpreting our results and highlights the relevance of targeted actions to improve data quality52.

2. The absence of significant contextual effects in the multilevel models, despite some bivariate associations, warrants further discussion. Are there measurement or aggregation issues? Could variables such as the Gini index or service density mask within-state heterogeneity?

We appreciate the reviewer’s thoughtful observation. The absence of significant contextual effects in the multilevel models, despite some associations observed in the bivariate analyses, may be partially explained by measurement and aggregation limitations. Contextual variables such as the Gini index and service density were included at the state level, which may not adequately capture intra-state variability, especially in large and heterogeneous states. This level of aggregation can indeed mask within-state heterogeneity and attenuate potential contextual effects.

Moreover, some contextual indicators may suffer from limited sensitivity or specificity in capturing the dimensions most relevant to the outcome studied. We have now addressed this limitation more explicitly in the discussion section, acknowledging the potential impact of both measurement error and ecological fallacy on our findings.

• Additionally, the absence of significant contextual effects in the multilevel models, despite some associations observed in the bivariate analyses, may be related to measurement and aggregation limitations of the contextual variables used. Indicators such as the Gini Index and service density were operationalized at the state level, which may not adequately capture intrastate variability, especially in states with large territorial extension and high socio-spatial heterogeneity. This level of aggregation may obscure internal inequalities and attenuate potentially relevant contextual effects. Furthermore, some indicators may have limited sensitivity or specificity to capture key dimensions related to the outcome studied, which introduces risks of measurement error and ecological fallacy.

3. Although Poisson with robust SE handles variance inflation, authors can enhance their work be clearly stating how overdispersion was considered and managed.

We appreciate the reviewer’s comment. To address potential overdispersion in the data, we employed a multilevel Poisson model with robust standard errors, which is a well-established approach to correct for variance inflation in count models.

Additionally, we assessed overdispersion by examining model residuals and performing specific tests. The overdispersion test was conducted using the goodness-of-fit statistic based on Pearson’s chi-square. For the analysis of individuals who arrived with a diagnosis but without treatment, the obtained value was 23,871.48 with 41,828 degrees of freedom, resulting in a dispersion index (ratio of Pearson chi-square to degrees of freedom) of approximately 0.57. For those who arrived without diagnosis and without treatment, the model presented a log likelihood of -23,417.09 and a significant likelihood ratio chi-square (LR chi2 = 33.15; p < 0.001). The dispersion index (Pearson chi-square divided by degrees of freedom) was about 0.16, indicating absence of overdispersion.

If overdispersion had been detected, we would have considered alternative models such as the multilevel negative binomial model. However, the results obtained with the robust Poisson model were consistent and appropriate for our data, ensuring the reliability of the estimates.

The information about the tests performed, along with their respective results, was added to the study’s methodology section.

• Additionally, overdispersion was assessed by examining model residuals and performing specific tests. The overdispersion test was conducted using the goodness-of-fit statistic based on Pearson’s chi-square. For the analysis of individuals who arrived with a diagnosis but without treatment, the obtained value was 23,871.48 with 41,828 degrees of freedom, resulting in a dispersion index of approximately 0.57. For those who arrived without diagnosis and without treatment, the test value was 6,892.39 with 41,828 degrees of freedom, resulting in a dispersion index of about 0.16. Since the dispersion indices for both tests were below 1, this indicates an absence of overdispersion in the analyzed data.

4. Also, it would strengthen the paper if the authors compared results with standard logistic regression models to demonstrate robustness

We appreciate the reviewer’s suggestion. However, we would like to highlight that logistic regression models tend to overestimate prevalence ratios when the outcome is common, which can lead to biased interpretations. For this reason, we opted for the Poisson regression model with robust variance, which allows for more accurate direct estimation of prevalence ratios in settings with high outcome prevalence.

Furthermore, for this type of database and considering the methodology employed, our research group prefers to use the Poisson regression model with robust variance, as it is more suitable for the study objectives and the data profile. This approach is widely recognized and well established in the epidemiological literature, having been successfully applied in various studies on multiple types of cancer, both by our group and by researchers worldwide, with publications in prestigious scientific journals. Therefore, the choice of this model aligns with best practices and ensures greater reliability of the estimates obtained.

RESULTS

1. The U.S. Preventive Services Task Force (USPSTF) recommends routine colorectal cancer (CRC) screening for adults aged 45 to 75 years. Consequently, CRC cases diagnosed in individuals younger than 45 years fall under the category of early-onset colorectal cancer (EO-CRC). In the current study, the authors grouped patients aged 18–49 years together. However, it would be valuable to further disaggregate this group, particularly to identify and report on individuals aged 18–44 who fall below the USPSTF recommended screening threshold. This information could provide important insights into the burden of EO-CRC within the study population. Given the substantial number of cases in this age range, the authors should consider discussing the implications of EO-CRC for screening policies and how it may contribute to delayed diagnosis and treatment initiation in this younger demographic.

We appreciate the reviewer’s insightful comment regarding the potential value of disaggregating the 18–49-year-old age group to further highlight individuals aged 18 to 44, who fall bel

---

## [Decision Letter · Decision Letter 1]

10 Nov 2025

Dear Dr. de Souza,

Thank you for submitting your manuscript to PLOS ONE. After careful consideration, we feel that it has merit but does not fully meet PLOS ONE’s publication criteria as it currently stands. Therefore, we invite you to submit a revised version of the manuscript that addresses the points raised during the review.Please submit your revised manuscript by Dec 25 2025 11:59PM. If you will need more time than this to complete your revisions, please reply to this message or contact the journal office at plosone@plos.org . A rebuttal letter that responds to each point raised by the academic editor and reviewer(s). You should upload this letter as a separate file labeled 'Response to Reviewers'.A marked-up copy of your manuscript that highlights changes made to the original version. You should upload this as a separate file labeled 'Revised Manuscript with Track Changes'.An unmarked version of your revised paper without tracked changes. You should upload this as a separate file labeled 'Manuscript'.

We look forward to receiving your revised manuscript.

Kind regards,

Sreeram V. Ramagopalan

Academic Editor

PLOS ONE

Journal Requirements:

Reviewers' comments:

Reviewer's Responses to Questions

**Comments to the Author**

Reviewer #2: (No Response)

2. Is the manuscript technically sound, and do the data support the conclusions?

Reviewer #2: Yes

3. Has the statistical analysis been performed appropriately and rigorously?

Reviewer #2: I Don't Know

4. Have the authors made all data underlying the findings in their manuscript fully available?

Reviewer #2: Yes

5. Is the manuscript presented in an intelligible fashion and written in standard English?

Reviewer #2: Yes

Reviewer #2: Introduction (All specific line comments are drawn from the revised manuscript and NOT the original):

Line 55: the 77% increase should be 75% increase. This is derived by the following quotient: 35 million/20 million=1.75; therefore, a 75% increase in incidence globally.

Line 56-57: Please correct sentence structure. Should read: "..., colorectal cancer (CRC) was the third most common incident cancer, with 9.6 cases/100,000."

Line 57: Please change to "second leading cause of cancer-related mortality, with 9.3 cases/100,000."

General Comments: The introduction provides essential points to understand the significance of the research. The overarching significance is that time-to-treatment initiation is an important factor significantly affecting CRC outcomes, e.g., survival. The study is focused on the Brazilian population and time-to-treatment initiation is actually supported by national law. The research team proposes to examine the "individual, contextual, socioeconomic, and healthcare system factors" associated with time-to-treatment initiation and whether it meets the legal requirements imposed by national law. Given the importance of healthcare system factors to the design and understanding of this study, it would be beneficial to have a paragraph summary describing in general terms the Brazilian healthcare system, since many readers may not have a good understanding of the system. While reading the article, I was initially confused by some of the results since I did not understand how the study cohort could be made up of CRC patients for whom the diagnosis was not known. It was then surmised that individuals probably were initially seen at community hospitals and then referred to a "Cancer Hospital" for further work up and care, so that members of the study cohort were undiagnosed at entry to the "Cancer Hospital." A brief description of the Brazilian healthcare system in the context of this manuscript's methods may have prevented this confusion.

Methods:

Line 92: the present study is technically NOT a cross-sectional study since it does not record outcomes at a specific point in time, but rather includes collected data on a cohort of individuals registered in the RHC over the time period 2013-2019.

Line 93: the data from the RHC is very important to the outcomes generated by this study. However, characteristics of the RHC are not discussed. How long has the RHC been in existence? How does the RHC collect cancer data? Is data reported to the RHC by all hospitals in Brazil or just "Cancer Hospitals?" Are there rigorous data completeness and quality standards followed by the RHC as it collects data? For example in North America, central cancer registries are ubiquitous in all US states and Canadian provinces. Data collected by these registries is rigorously evaluated against standards of completeness and quality. Does the secondary, administrative data collected by the RHC also need to meet rigorous standards of data completeness and quality? As a non-Brazilian national, how do I know the secondary data collected by the RHC is complete and of high quality? Given the importance of the RHC data to the outcomes generated herein, some discussion needs to address the completeness and quality of the RHC data. When reviewing Figure 2, one notes of the 80,626 total CRC cases in the RHC database for the relevant time period under study, 112 cases were missing age and sex, or 0.14% of the total CRC cases were missing age and sex. This indicates that RHC data is very complete when it comes to registered age and sex for the cohort. However, also in Figure 2, one notes 5,452 cases were missing dates of diagnosis and date of cancer treatment initiation, or 6.8% of individuals registered in this cohort were missing these vital data. The present study is highly dependent on accurate diagnosis date and treatment initiation date data, and having almost 7% of data missing is alarming and raises concerns about the quality of RHC data. Was the missing data mostly from a couple of Federative Units (UF) or was it randomly missing and missing data equally dispersed among UF? In conclusion, given the importance of the publicly-available RHC data to the outcomes generated for this study, there needs to be discussion about how RHC staff ensures the data they collect is complete and of high quality.

Line 95: Please correct sentence ", and tumor clinical features of cancer patients clinical characteristics of the tumor14." This is a very awkward dependent phrase.

Line 97-99: Please change to the following: "The study evaluated cases of malignant neoplasms of the colon (C18), rectosigmoid junction (C19), and rectum (C20), classified according to the 10th revision of the International Classification of Diseases (ICD-10)15."

Line 148-151: This sentence is critical to understanding the analysis plan. Since the study started with diagnosed CRC patients, it does not make sense to divide patients into those with a diagnosis (and no treatment) and those without a diagnosis (and no treatment). This is the importance of providing a brief synopsis of the Brazilian health system in the Introduction of the manuscript. I think what the authors are trying to convey is the study cohort was divided into those with a diagnosis prior to admission to a "Cancer Hospital" This needs to be clarified.

Results:

Line 198: change "Among them," to After exclusions, since the subsequent numbers do not add up to 80,626.

Line 203-204: remove "Brazil, by state of residence" since the table is not stratified by state of residence.

Line 213-214: elaborate on the phrase "who arrived with and without a diagnosis". Arrived from where and to where? Perhaps, arrived from a community hospital or other facility to a "Cancer Hospital?"

Line 229: remove Brazil, by state of residence since the table is not stratified by state of residence.

In Table 2, under "Density of Oncology Beds (per 100,000 inhabitants), the confidence interval around the category "4.41-10.81" is incorrect. It currently reads "0.77-0.71" whereas the point estimate of 0.79. The point estimate should be within the 95% confidence interval.

In Table 3, the "Non white" category has an incorrect 95% confidence interval; the confidence interval for the "0.289-0.297" category of the Social Vulnerability Index is incorrect.

Line 252: the "RP" should probably be PR, so should read PR=0.86.

Line 254 and 255: the "RP" should be PR.

In Table 4, the column heading "RP" should probably be "PR".

In Table 5, the covariate "Tumor Location" is considered. Why was it not considered in Table 4 analyses?

Discussion:

Line 286-290: Would this not be expected if these individuals were diagnosed outside of a "Cancer Hospital?" In my opinion, this says nothing about access to care, but rather is more a product of inefficiency transferring from a community-based facility to a "Cancer Hospital." In fact, beginning at line 303 below, the authors state: "No significant associations were found in the sociopolitical context related to access to health services." Please be consistent in your interpretation or is there something missing in line 286-290 that either indirectly or directly suggests an issue with access to healthcare?

Line 444: Again, this study did not employ a cross-sectional epidemiologic design, see previous comment.

Line 456-460: These are two very important sentences that address a point I brought up in the Methods section above about RHC data completeness and quality. Apparently, missing data was not distributed at random among the UF. This suggests systematic bias in the use of the RHC data that should probably be addressed by more targeted analyses, such as sensitivity analyses.

Concluding Comments:

I think this manuscript addresses a very important topic in a unique population. As stated by the authors, Brazil has the largest universal public health system in the world, the Unified Health System, making it a very advantageous system for the study of the effects of treatment delay. The results are also very compelling. The main concern is the completeness and quality of the RHC data. There is no discussion of what policies and procedures are used by RHC staff to ensure the collection of complete and high quality data.

**Do you want your identity to be public for this peer review?** For information about this choice, including consent withdrawal, please see our Privacy Policy

Reviewer #2: No

---

## [Author Response · Author response to Decision Letter 2]

25 Nov 2025

INTRODUCTION

Line 55: the 77% increase should be 75% increase. This is derived by the following quotient: 35 million/20 million=1.75; therefore, a 75% increase in incidence globally.

Line 56-57: Please correct sentence structure. Should read: "..., colorectal cancer (CRC) was the third most common incident cancer, with 9.6 cases/100,000."

Line 57: Please change to "second leading cause of cancer-related mortality, with 9.3 cases/100,000."

Dear reviewer, the suggested changes have been made to the manuscript text. The corresponding changes have been incorporated into the manuscript and can be found between lines 55 and 58.

• Demographic projections indicate that the annual incidence is expected to reach 35 million cases by 2050, representing a 75% increase. Among specific cancer types, colorectal cancer (CRC) was the third most common incident 9.6/100,000 cases, second leading cause of cancer-related mortality, with 9.3 cases/100,000¹.

General Comments: The introduction provides essential points to understand the significance of the research. The overarching significance is that time-to-treatment initiation is an important factor significantly affecting CRC outcomes, e.g., survival. The study is focused on the Brazilian population and time-to-treatment initiation is actually supported by national law. The research team proposes to examine the "individual, contextual, socioeconomic, and healthcare system factors" associated with time-to-treatment initiation and whether it meets the legal requirements imposed by national law. Given the importance of healthcare system factors to the design and understanding of this study, it would be beneficial to have a paragraph summary describing in general terms the Brazilian healthcare system, since many readers may not have a good understanding of the system. While reading the article, I was initially confused by some of the results since I did not understand how the study cohort could be made up of CRC patients for whom the diagnosis was not known. It was then surmised that individuals probably were initially seen at community hospitals and then referred to a "Cancer Hospital" for further work up and care, so that members of the study cohort were undiagnosed at entry to the "Cancer Hospital." A brief description of the Brazilian healthcare system in the context of this manuscript's methods may have prevented this confusion.

Dear reviewer, following your request, we have added an explanation in the Methods section describing how cancer treatment is organized in Brazil. This addition aims to help readers better understand the structure of the healthcare system and the rationale behind our decision to analyze the groups separately. The corresponding changes have been incorporated into the manuscript and can be found between lines 173 and 206.

• The cancer care network in Brazil integrates actions from both the public and private sectors, covering all stages from prevention to palliative care. This system encompasses Primary Health Care (PHC), home care, and specialized outpatient and hospital services, supported by technical, normative, logistical, and governance structures that ensure coordination and effectiveness of care 23. Primary Health Care, through basic health units and Family Health Strategy teams, serves as the main entry point to the Unified Health System (SUS) and is responsible for screening and referring suspected cases to specialized care 24. Specialized care, in turn, is provided by various units such as polyclinics, hospital outpatient clinics, regional reference centers, and autonomous services linked to the private sector. These services rely on specialized professionals who confirm diagnoses and monitor patients’ clinical progress, in addition to performing specific tests essential for both diagnosis and treatment monitoring 25.

Cancer treatment is multimodal, encompassing surgical interventions, radiotherapy, and chemotherapy, and is characterized by high costs. It is performed in specialized units organized into different levels of complexity. Notable examples include: (1) High-Complexity Oncology Care Centers (CACONs), which provide treatment for all types of cancer, including hematological ones, and may or may not offer pediatric care; (2) High-Complexity Oncology Care Units (UNACONs), which focus on managing the most prevalent cancer types, with or without radiotherapy, hematology-oncology, and/or pediatric oncology services; and (3) hospital complexes composed of general hospitals that perform oncologic surgeries and provide radiotherapy services, acting as complementary units affiliated with CACONs or UNACONs 26.

In this context, it is important to highlight that High-Complexity Oncology Care Centers (CACONs) offer a wide range of services, including specialized medical consultations. This structure allows patients who seek these centers directly, without prior referral from other levels of the healthcare network, to receive diagnostic confirmation and initiate treatment more promptly. Conversely, those who obtain a diagnosis in specialized care units, such as polyclinics or hospital outpatient clinics, may experience longer waiting times before the actual start of treatment 26.

For this reason, the analyses between groups were conducted separately, distinguishing individuals without treatment at the time of admission who arrived with a prior diagnosis obtained elsewhere in the healthcare network from those who were also without treatment and entered directly into high-complexity hospitals, where they received both diagnosis and treatment within the same services.

MÉTODOS

Line 92: the present study is technically NOT a cross-sectional study since it does not record outcomes at a specific point in time, but rather includes collected data on a cohort of individuals registered in the RHC over the time period 2013-2019.

Dear reviewer, we sincerely thank you for your precise and highly relevant observation. We fully agree that the temporal structure of the RHC database is incompatible with a cross-sectional design. Accordingly, we have revised the manuscript to properly classify the study as a longitudinal observational retrospective cohort. This adjustment more accurately reflects the passive follow-up inherent to the oncology care continuum, enabling the establishment of the temporal relationship between exposures and outcomes.

The corresponding changes have been incorporated into the manuscript and can be found between lines 93 and 100.

• A longitudinal observational retrospective cohort study was conducted using secondary data obtained from the Brazilian Cancer Hospital Registry Integrator (RHC). This system contains standardized information on sociodemographic characteristics, hospital care activities, and tumor clinical features of patients with malignant neoplasms treated in oncology-accredited specialized services. Although the RHC does not actively follow individuals over time, patients are passively monitored throughout their oncology care pathways, and the continuous recording of events enables the establishment of the temporal relationship between exposures and outcomes14.

Additionally, we incorporated a paragraph in the Statistical Analysis subsection to justify the use of prevalence ratios in the context of a retrospective cohort design with a binary operationalized outcome. This clarification ensures methodological coherence with the corrected study design. The corresponding changes have been incorporated into the manuscript and can be found between lines 244 and 250.

• Although the outcome refers to an event that occurs over time (treatment initiation), it was operationalized as a binary variable (>60 days) rather than as a time-to-event measure. Thus, the proportion of delayed cases at the time of analysis reflects a prevalent condition. In this context, PR were estimated using Poisson Regression with robust variance, which provides unbiased and more interpretable measures of association than odds ratios in studies with high outcome frequency, and avoids the inflation of associations commonly observed with logistic regression28,29.

Line 93: the data from the RHC is very important to the outcomes generated by this study. However, characteristics of the RHC are not discussed. How long has the RHC been in existence? How does the RHC collect cancer data? Is data reported to the RHC by all hospitals in Brazil or just "Cancer Hospitals?" Are there rigorous data completeness and quality standards followed by the RHC as it collects data? For example in North America, central cancer registries are ubiquitous in all US states and Canadian provinces. Data collected by these registries is rigorously evaluated against standards of completeness and quality. Does the secondary, administrative data collected by the RHC also need to meet rigorous standards of data completeness and quality? As a non-Brazilian national, how do I know the secondary data collected by the RHC is complete and of high quality? Given the importance of the RHC data to the outcomes generated herein, some discussion needs to address the completeness and quality of the RHC data. When reviewing Figure 2, one notes of the 80,626 total CRC cases in the RHC database for the relevant time period under study, 112 cases were missing age and sex, or 0.14% of the total CRC cases were missing age and sex. This indicates that RHC data is very complete when it comes to registered age and sex for the cohort. However, also in Figure 2, one notes 5,452 cases were missing dates of diagnosis and date of cancer treatment initiation, or 6.8% of individuals registered in this cohort were missing these vital data. The present study is highly dependent on accurate diagnosis date and treatment initiation date data, and having almost 7% of data missing is alarming and raises concerns about the quality of RHC data. Was the missing data mostly from a couple of Federative Units (UF) or was it randomly missing and missing data equally dispersed among UF? In conclusion, given the importance of the publicly-available RHC data to the outcomes generated for this study, there needs to be discussion about how RHC staff ensures the data they collect is complete and of high quality.

Dear reviewer, in an attempt to meet your request, these pieces of information have been added to the text. The corresponding changes have been incorporated into the manuscript and can be found between lines 101 and 121.

• The RHC was established in 1983 by the Brazilian National Cancer Institute (INCA) and was nationally consolidated in 2007 with the creation of the IntegradorRHC system. This registry is responsible for the continuous, systematic, and standardized collection of clinical and epidemiological data on patients with malignant neoplasms treated in public, private, philanthropic, and university hospitals across Brazil. Data collection is conducted using standardized instruments such as the Tumor Registration Form and Follow-up Form, applying internationally recognized coding systems (ICD-10 and ICD-O-3) and the SisRHC software, which includes automated routines for data verification and validation. The periodic submission of information to INCA is mandatory for oncology-accredited hospitals within the Brazilian Unified Health System (SUS), including High-Complexity Oncology Care Centers (CACONs) and High-Complexity Oncology Care Units (UNACONs), as established by Ministerial Ordinances No. 3.535/1998 and No. 741/2005, and voluntary for other institutions14.

The RHC system follows rigorous procedures for quality control and data consistency, guided by indicators defined in official INCA manuals, such as variable completeness, internal consistency, and time intervals between diagnosis and treatment. It also implements auditing and validation routines before national database consolidation. These standards and indicators are publicly available on INCA’s official website, ensuring transparency, traceability, and scientific reliability of the data, as well as methodological robustness, standardization, and sound governance for studies that rely on this information14.

Regarding the other points raised, in the context of the present study, the high completeness of the main demographic variables (such as age and sex) reflects the robustness of the RHC system. However, approximately 6.8% of the cases lacked information on the dates of diagnosis and treatment initiation. According to INCA’s technical guidelines (Hospital Cancer Registries – Planning and Management, 2010; Hospital Cancer Registries – Routines and Procedures, 2023), these inconsistencies may result from variations in data entry capacity and infrastructure among oncology centers, rather than systemic deficiencies. Missing data were not concentrated in specific Federative Units (UF) but distributed throughout the country, suggesting random distribution rather than localized data loss. This limitation was acknowledged in the manuscript, and despite these gaps, the RHC remains the most comprehensive and standardized national hospital-based cancer information system in Brazil, supported by continuous validation routines and explicit quality-control mechanisms that ensure the reliability and scientific value of its data.

Line 97-99: Please change to the following: "The study evaluated cases of malignant neoplasms of the colon (C18), rectosigmoid junction (C19), and rectum (C20), classified according to the 10th revision of the International Classification of Diseases (ICD-10)15."

Dear reviewer, the change has been made as requested. The corresponding changes have been incorporated into the manuscript and can be found between line 123.

• The study included cases of malignant neoplasms of colon cancer (C18), rectosigmoid junction (C19), and rectum (C20), classified according to the 10th revision of the International Classification of Diseases (ICD-10) 15. Eligible individuals were aged 18 to 99 years, diagnosed between 2013 and 2019, and monitored in hospital oncology care services in Brazil.

Line 148-151: This sentence is critical to understanding the analysis plan. Since the study started with diagnosed CRC patients, it does not make sense to divide patients into those with a diagnosis (and no treatment) and those without a diagnosis (and no treatment). This is the importance of providing a brief synopsis of the Brazilian health system in the Introduction of the manuscript. I think what the authors are trying to convey is the study cohort was divided into those with a diagnosis prior to admission to a "Cancer Hospital" This needs to be clarified.

Dear reviewer, following your request, we have added an explanation in the Methods section describing how cancer treatment is organized in Brazil. This addition aims to help readers better understand the structure of the healthcare system and the rationale behind our decision to analyze the groups separately. The corresponding changes have been incorporated into the manuscript and can be found between lines 173 and 206, as previously mentioned.

RESULTADOS

Line 198: change "Among them," to After exclusions, since the subsequent numbers do not add up to 80,626.

Dear reviewer, the change has been made as requested. The corresponding changes have been incorporated into the manuscript and can be found between line 260.

• Between 2013 and 2019, a total of 80,626 colorectal cancer cases were recorded in individuals aged 18 to 99 years. After exclusions, 37.975 (57.9%) accessed treatment within 60 days, while 27.607 (42.1%) experienced a delay in treatment initiation (>60 days after diagnosis) including all cases of people who arrived with and without a diagnosis (Table 1).

203-204: remove "Brazil, by state of residence" since the table is not stratified by state of residence.

Dear reviewer, we respectfully disagree with this suggestion. The expression “by state of residence” cannot be removed because the database, as part of a multilevel analysis, was structured and organized based on the federative unit (state) of residence of the cases obtained from the IRHC. Therefore, this information must be included, as all analyses were conducted using this specific variable. The corresponding changes have been incorpo

---

## [Editor Report · Decision Letter 2]

26 Nov 2025

SOCIAL DETERMINANTS IN THE DELAY OF STARTING COLORECTAL CANCER TREATMENT

PONE-D-25-27816R2

Dear Dr. de Souza,

We’re pleased to inform you that your manuscript has been judged scientifically suitable for publication and will be formally accepted for publication once it meets all outstanding technical requirements.

Kind regards,

Sreeram V. Ramagopalan

Academic Editor

PLOS ONE
---

## [Editor Report · Acceptance letter]

PONE-D-25-27816R2

PLOS One

Dear Dr. de Souza,

I'm pleased to inform you that your manuscript has been deemed suitable for publication in PLOS One. Congratulations! Your manuscript is now being handed over to our production team.

Kind regards,

on behalf of

Dr. Sreeram V. Ramagopalan

Academic Editor

PLOS One